# OmniKV: Dynamic Context Selection for Efficient Long-Context LLMs

**Jitai Hao**[2*‡]**, Yuke Zhu**[1*]**, Tian Wang**[1]**, Jun Yu**[3]**, Xin Xin**[2]**, Bo Zheng**[1]**, Zhaochun Ren**[4]**, Sheng Guo**[†1]

[1]MYbank, Ant Group   [2]Shandong University   [3]Harbin Institute of Technology
[4]Leiden University   {felix.yk,guosheng.guosheng}@mybank.cn
jitaihao@mail.sdu.edu.cn,z.ren@liacs.leidenuniv.nl

## Abstract

During the inference phase of Large Language Models (LLMs) with long context, a substantial portion of GPU memory is allocated to the KV cache, with memory usage increasing as the sequence length grows. To mitigate the GPU memory footprint associate with KV cache, some previous studies have discarded less important tokens based on the sparsity identified in attention scores in long context scenarios. However, we argue that attention scores cannot indicate the future importance of tokens in subsequent generation iterations, because attention scores are calculated based on current hidden states. Therefore, we propose OmniKV, a token-dropping-free and training-free inference method, which achieves a 1.68x speedup without any loss in performance. It is well-suited for offloading, significantly reducing KV cache memory usage by up to 75% with it. The core innovative insight of OmniKV is: Within a single generation iteration, there is a high degree of similarity in the important tokens identified across consecutive layers. Extensive experiments demonstrate that OmniKV achieves state-of-the-art performance across multiple benchmarks, with particularly advantages in chain-of-thoughts scenarios. OmniKV extends the maximum context length supported by a single A100 for Llama-3-8B from 128K to 450K. Our code is available at https://github.com/antgroup/OmniKV.git

## 1 Introduction

Large Language Models (LLMs) have demonstrated their profound impact across a multitude of applications, such as chatbots (Achiam et al., 2023; Meta, 2024), agents (Wu et al., 2023; Chan et al., 2023; Liu et al., 2023b), and embodied robotics (Mai et al., 2023; Zhang et al., 2024b). These applications confirm the demand for LLMs with strong long context processing capabilities to tackle complex tasks based on prior interaction histories or given information materials. Currently, some studies have successfully extended the maximum context length of LLMs (Peng et al., 2023; Yang et al., 2024; Young et al., 2024; Meta, 2024; Chen et al., 2023).

In long-context scenarios, LLMs inference incurs more massive GPU memory usage. Particularly, a significant portion of this is attributed to the KV cache, which is proposed to accelerate LLMs generation and reduce redundant computation. Moreover, the memory occupied by the KV cache increases linearly with the sequence length. For instance, in the case of Llama-3-8B, with a batch size of 8 and a context length of 128K tokens, the KV cache alone occupies over 134GB of GPU memory, presenting a significant challenge to inference systems. To alleviate the high memory usage of the KV cache in long-context scenarios, previous studies (Zhang et al., 2024c; Li et al., 2024; Liu et al., 2024a) have attempted to identify and discard **less important** tokens based on sparsity in attention in long-context scenarios (Wang et al., 2021; Ribar et al., 2024). Tokens with lower cumulative attention scores are discarded, thereby reducing GPU memory occupation.

However, we argue that in multi-step reasoning scenarios, the **important** tokens vary depending on the reasoning step. This variation arises because attention scores are calculated based on the

---

* Equal contribution. †Corresponding author. ‡Work done during an internship at MYbank.

current hidden states, meaning that the attention scores of tokens only reflect their relevance to current reasoning step. Consequently, some tokens with low scores may be recalled as important and relevant tokens in subsequent reasoning steps as shown in Figure 1b. Discarding tokens may lead to the loss of crucial information for completing subsequent reasoning steps.

Motivated by this, we propose OmniKV, a novel inference method that retains the KV cache for all tokens. This approach not only achieves performance comparable to that of the original model but also accelerates decoding efficiency when context is longer than 32K. OmniKV is mainly built upon an innovative insight: For a specific context, the tokens with high attention scores are very similar across various layers. We refer to this phenomenon as **Inter-Layer Attention Similarity**. In other words, a similar set of tokens is identified as important across multiple layers.

Specifically, OmniKV offloads most layers' KV cache to CPU memory when processing the input prompt (i.e., the prefill stage) but retains few "**filter**" layers' entire KV cache. During the next token prediction (i.e., the decode stage), OmniKV first exploits the sparsity of attention map, selecting a few top-scoring tokens using top-k in these few "filter" layers. Next, other layers directly use the subset of tokens chosen by the preceding filter layers as context. This way, OmniKV only needs to load a small subset of tokens from CPU memory into GPU memory for attention computation. For a decoding iteration, since many layers share the same index of tokens, we only perform $\leq 3$ transfers between GPU and CPU. By using asynchronous transfer to overlap computation with transfer time, the efficiency of decoding is accelerated due to the shorter context computed in most layers.

We conducted extensive experiments on various LLMs, including Llama-3-8B-262K [*], Yi-9B-200K (Young et al., 2024) and Llama-3.1-70B (Meta, 2024) across benchmarks LongBench (Bai et al., 2023) and InfiniteBench (Zhang et al., 2024a). We tested both single-step reasoning and multi-step reasoning (Chain of Thoughts) setups. The results showed that OmniKV achieved the best performance in both settings, especially in the multi-step reasoning setup. This demonstrates the necessity of drop-free and the effectiveness of dynamic context selection. Moreover, OmniKV can accelerate inference when context lengths exceed 32K. By using only a single A100 GPU, compared to the original model's efficiency, OmniKV achieves a 1.7x speedup with 128K context. When using three A100s for origin model to pipeline inference with 450K context, OmniKV still achieves a 1.87x speedup while using only a single A100.

## 2 RELATED WORK

**Token Dropping and Offloading.** Most similar studies to ours focus on discarding unimportant tokens based on accumulated attention scores after the prefill stage. These tokens are then completely dropped in the decode stage (Zhang et al., 2024c; Liu et al., 2024a; Li et al., 2024; Ge et al., 2024a). However, these methods might discard tokens that could become important in future reasoning steps. To ensure the lossless of information, we dynamically select a sparse subset of the KV cache for each generation iteration to guarantee performance. In a related approach, Quest (Tang et al., 2024) recognizes the importance of dynamic selection. Nonetheless, it fails to reduce memory usage and may compromise recall. Due to representing a block with a single vector, Quest's capacity to retrieve relevant tokens may be compromised.

Another type of similar works involves KV cache offloading. Many studies offload layers' KV cache to CPU memory when VRAM is not sufficient (Sheng et al., 2023; Kwon et al., 2023b). However, these methods do not leverage sparsity in attention. The data transfer volume over PCIe is 10 times more than ours in long-context scenarios. InfLLM (Xiao et al., 2024a) similarly divides sequences into blocks, selecting a few representative vectors as retrieval keys for a block. Then offloads other data to CPU memory. However, the chosen few representative vectors may fail to fully capture the block's information, resulting in a relatively low recall rate.

**Other Efficient Methods.** Considerable efforts have been made to minimize KV cache while incurring minimal performance loss to the model. One class of work compresses KV cache, such as ICAE (Ge et al., 2024b) and Gist (Mu et al., 2024), which utilize LLMs as auto-encoders to compress the context to a shorter sequence. Additionally, there are works that directly compress prompts at the language level, thereby indirectly compressing KV cache, such as LLMLingua (Jiang et al., 2023).

---

[*]https://huggingface.co/gradientai

Similar to model weight quantization, there are also attempts to quantize KV cache, such as KIVI (Liu et al., 2024b) and SmoothQuant (Xiao et al., 2023). These compression or quantization works are orthogonal to our method and can be used in conjunction.

**Sparsity in LLMs.** The sparsity of attention in long-context scenarios has been observed by previous studies (Liu et al., 2023a; Ribar et al., 2024; Wang et al., 2021). For example, Minference (Jiang et al., 2024b) has demonstrated that with a context of 128k, only 4k tokens are required to accumulate 96.4% of the total attention score. However, the tokens with high attention scores vary across different generation iterations, indicating that the sparse pattern is dynamic (Tang et al., 2024). This implies that it is necessary to compute the full attention in every generation iteration and every layer to determine the sparse pattern. Quest (Tang et al., 2024) and SparQ (Ribar et al., 2024) employed approximate attention methods to circumvent the computationally expensive full attention. Infini-Gen (Lee et al., 2024) introduced cross-layer similarity between consecutive two layers, leveraging this characteristic to pre-select the critical KV cache. However, the loading time may still exceed the computation time, resulting in GPU idleness. In contrast, we observe that the sparse patterns between different layers exhibit high similarity, and thus we only compute full attention for a few layers to obtain the sparse patterns for subsequent layers. To the best of our knowledge, we are the first to highlight this.

# 3 INSIGHTS

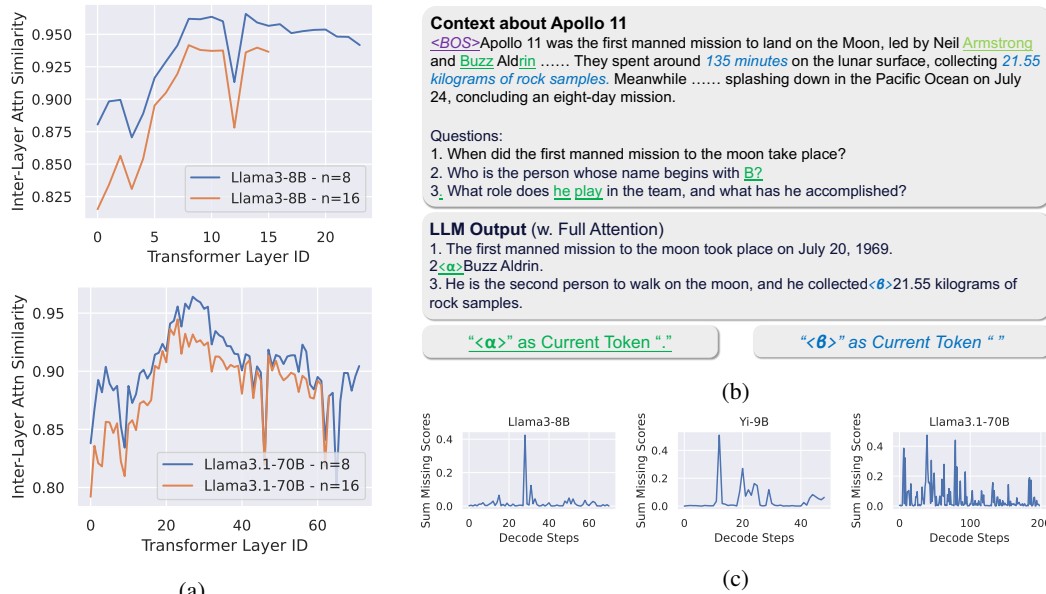

Figure 1: Analysis of attention. (a) Inter-Layer Attention Similarity. This shows a high similarity of important tokens/sub-context between layers even 16 layers apart. (b) An example of a multihop question demonstrates the variation in important tokens across different generation iterations. (c) Analysis of variety in important tokens. We retained the set of important tokens, then calculated the cumulative attention scores of missing tokens.

Our work is grounded in three pivotal insights, which we verify through experiments utilizing popular models Llama-3-8B-Instruct, Yi-9B-200k, and Llama-3.1-70B-Instruct.

**Intra-Layer Attention Sparsity.** Studies have consistently revealed that attention matrices within LLMs layers exhibit sparsity (Wang et al., 2021; Deng et al., 2024). This characteristic implies that LLMs can generate nearly equivalent outputs by focusing on a reduced subset of tokens. Some studies have enhanced inference speed or reduced GPU memory requirements based on sparsity (Zhang et al., 2024c; Li et al., 2024; Tang et al., 2024). Due to the presence of sparsity, OmniKV

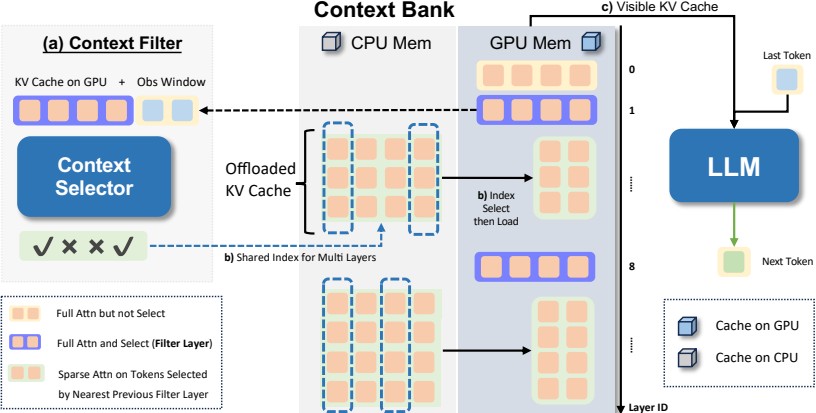

Figure 2: Overall Framework of OmniKV in decode stage. There are three types of layers in Om-niKV. In **prefill** stage, all layers perform full attention and generate KV cache of context. Then OmniKV offloads the KV cache generated by green layers to CPU memory. In the **decode** stage, orange layers perform full attention because of lower sparsity or inference efficiency. Purple (filter) layers perform full attention and **a)** select important tokens using context selector based on attention scores calculated over observation window. Green layers **b)** load the subset of KV cache selected by preceding filter layers to GPU and perform sparse attention. **c)** Only the KV Cache on the GPU is visible to the LLM for generating next token.

utilizes only a small subset of tokens in most layers. In this way, we not only reduced computation but also decreased the communication volume between CPU and GPU.

**Inter-Layer Attention Similarity.** We introduce the concept of inter-layer attention similarity, which is defined as, a fixed subset of tokens that receive significant attention in a specific layer, maintains their prominence throughout successive layers. The value of similarity of a layer can also be viewed as "**filter**" ability of this layer, and is calculated as the mean of the summation of fixed tokens subset's attention scores in subsequent layers. Figure 1a demonstrates that, beyond a certain shallow layer, the similarity for one layer to successive $n$ layers becomes exceptionally high. Although the overall similarity between layers is already high, some layers exhibit higher "filter" ability than others. We refer to these layers as "filter" layers. Subsequently, these layers function as context selectors within OmniKV, identifying crucial tokens for each generation iteration and thus facilitating sparse attention in subsequent layers.

**Inter-Token Attention Variability.** Intuitively, the important tokens should vary throughout the generative process of LLMs, particularly under multi-task or multi-reasoning scenarios such as the *Chain of Thoughts* (CoT) (Wei et al., 2023). As shown in Figure 1b, for a multi-hop question, we highlight 12 tokens with the highest attention scores for two decoding steps respectively in the CoT scenario. We can observe that, apart from the special *BOS* token, the other important tokens are **entirely different**. Meanwhile, as demonstrated in Figure 1c, empirical studies on the Multi-Hop QA task (Ho et al., 2020) also confirm this variability. For each generation iteration, we compute the cumulative attention scores for the missing tokens to a token set, which stores $25\%$ most pivotal tokens during the prefill stage (i.e. Heavy Hitters in H2O). The **spikes** observed in the figure indicate that some missing tokens have significant attention scores. This phenomenon substantiates our intuition that the subsets of critical tokens identified exhibit significant fluctuations across different generative steps. Motivated by this insight, OmniKV retains all KV cache to ensure that performance remains unaffected.

## 4 METHOD

Based on the aforementioned insights, we propose OmniKV, a token-dropping-free and training-free inference method. This design empowers OmniKV to sustain the performance of LLMs in multi-

reasoning settings. As depicted in Figure 2, OmniKV comprises two pivotal modules: the **Context Bank** and the **Context Selector**.

The inference of auto-regressive LLMs can be divided into two stages: 1) *Prefill*, which encodes the intermediate computational state of the input prompt as KV cache $\mathbf{K}, \mathbf{V}$ to circumvent redundant KV vector computations, and outputs the next token as the first input for decoding; 2) *Decode*, which takes the token predicted from the previous decode iteration as the current token, and predicts the next token.

During the prefill stage, OmniKV initialize the **Context Bank** to store most "non-filter" layers' KV cache in CPU memory based on inter-layer attention similarity. In the decode phase, OmniKV adopts a plug-and-play **Context Selector** that dynamically identifies important subsets of KV cache $\mathbf{K}, \mathbf{V}$ on a few "**filter**" layers. Then the Context Bank propagates the selections to the "non-filter" layers and load the corresponding subset of KV cache in a pack to GPU memory, since these layers share a same index of tokens. In this way, OmniKV reduces computation and data transfer costs.

## 4.1 Context Bank

The proposed Context Bank utilizes inter-layer attention similarity to prefetch important tokens. In scenarios of insufficient GPU memory, the Context Bank can also asynchronously preload the corresponding KV cache from CPU memory, thereby alleviating memory constraints.

For the sake of simplicity in analysis, we will ignore the batch size here. In the prefill phase, an $L$-layer LLM creates KV caches, $\{\mathbf{K}_i, \mathbf{V}_i\}_{i=1}^{L}$, by applying attention projection matrices $\mathbf{W}_i^k$ and $\mathbf{W}_i^v$ to hidden states $\mathbf{h}_i^p$, yielding tensors in $\mathbb{R}^{H \times N \times d}$ for both keys and values. Here, $N$ represents the length of the tokenized prompt $p$ within the prefill context, $H$ signifies number of attention heads, and $d$ denotes the hidden size of per attention head.

Firstly, We need to determine which layers are more effective for identifying important tokens. These selected layers are referred to as "**filter**" layers. As illustrated in Figure 1a, although the similarity between adjacent 16 layers is high, a significant gap is observed between the mean similarities of 8 adjacent layers compared to 16 layers. To enhance performance by reducing the distance between "filter" layers, OmniKV utilizes a set of hyperparameters $\mathbb{L}$, where the size $(m, m \leq 3)$. Compared to using a single "filter" layer, the sub-context used by "non-filter" layers theoretically exhibits a higher degree of similarity when using multiple "filter" layers.

Consequently, OmniKV performs full attention on the layers within $\mathbb{L}$ to identify a small subset of important tokens $\mathbf{T}_i, (i \in \mathbb{L})$. Due to the shallow layers' reduced sparsity, OmniKV also performs full attention without selection on preceding layers up to layer $\mathbb{L}_0, (l < \mathbb{L}_0)$. Then, OmniKV utilizes **only** these important tokens $\mathbf{T}_i$ as sub-context for the sparse attention layers $l$, where $\mathbb{L}_i < l < \mathbb{L}_{i+1}$. Here, $\mathbf{h}_i^w$ represents the hidden states of the observation window at layer $i$. The context selector identifies important tokens that have significant attention scores over the observation window.

$$\mathbf{T}_i = \begin{cases} \text{ContextSelector}(\mathbf{h}_i^w, \mathbf{K}_i) & \text{if } i \in \mathbb{L} \\ \mathbf{T}_{i-1} & \text{otherwise} \end{cases} \quad \text{for } i \geq \mathbb{L}_0 \tag{1}$$

To avoid unnecessary GPU waiting when loading KV cache, OmniKV also performs full attention on the $\mathbb{L}$-adjacent layers, denoted as $\{l+1\}_{l \in \mathbb{L}}$. This interleaves data transfer with computation. Here, $\mathbf{h}_i^l$ represents the hidden states of the last token. Finally, the entire attention mechanism can be formulated as follows:

$$\text{out}_i = \begin{cases} \text{Attention}_i(\mathbf{h}_i^l, \mathbf{K}_i, \mathbf{V}_i) & \text{if } i \in \mathbb{L} \text{ or } i-1 \in \mathbb{L} \text{ or } i < \mathbb{L}_0 \\ \text{Attention}_i(\mathbf{h}_i^l, \mathbf{K}_i[\mathbf{T}_i], \mathbf{V}_i[\mathbf{T}_i]) & \text{otherwise} \end{cases} \tag{2}$$

OmniKV significantly reduces the sequence length to less than $10\%$ in sparse attention layers, which leads to a decrease in time complexity. Upon identifying the critical tokens $\mathbf{T}_i$ in $\mathbb{L}_t$ (where $\mathbb{L}_t = i$), OmniKV retrieves the corresponding subset of KV caches $\mathbf{K}_j[\mathbf{T}_i], \mathbf{V}_j[\mathbf{T}_i]$ (where $\mathbb{L}_t + 1 < j < \mathbb{L}_{t+1}$) for sparse layers as a sub-context from the CPU memory.

**Packed Load.** Since layers between "filter" layers share the same sub-context tokens' index $\mathbf{T}$, the KV cache for a series of consecutive sparse attention layers can be packed and loaded from the CPU to the GPU at the nearest preceding "filter" layer. Consequently, OmniKV conducts only

$m, m \leq 3$ loads, significantly reducing the slow PCIe transfer overheads compared to loading at each layer.

## 4.2 CONTEXT SELECTOR

As described in Section 4.1, OmniKV selects important tokens $\mathbf{T}_i$ in "filter" layers $\mathbb{L}$. Inspired by previous works (Li et al., 2024; Xiao et al., 2024a), we propose a unified framework for token selection. OmniKV selects important tokens based on a score vector $\mathbf{S}_i \in \mathbb{R}^N$. The score $\mathbf{S}_i$ is calculated using a observation window $\mathbf{h}_i^w$. Commencing with the local window as the query states, and the full context $\mathbf{h}_i^c$ as the key states, we compute the attention scores $\mathbf{A}_i$:

$$\mathbf{Q}_i = \mathbf{W}_i^q \mathbf{h}_i^w, \quad \mathbf{K}_i = \mathbf{W}_i^k \mathbf{h}_i^c, \qquad \mathbf{A}_i = \text{Softmax}\left(\frac{\mathbf{Q}_i \mathbf{K}_i^\top}{\sqrt{d}}\right), \quad \mathbf{A}_i \in \mathbb{R}^{H \times |\mathbf{h}_i^w| \times |\mathbf{h}_i^c|} \qquad (3)$$

Next, to get the score $\mathbf{S}_i$, we first apply reduce-max to obtain the maximum score across attention heads. Subsequently, a weighted vector $\alpha$ is utilized to perform a weighted summation on the attention scores. Finally, we leverage $\texttt{topk}$ to identify the important tokens $\mathbf{T}_i$ on score $\mathbf{S}_i$.

$$\mathbf{S}_i = \sum_{j=0}^{|\mathbf{h}_i^w|-1} \alpha_j \max_{0 \leq h < H} \mathbf{A}_i[h, j], \quad \mathbf{S}_i \in \mathbb{R}^{|\mathbf{h}_i^c|}, \qquad \mathbf{T}_i = \underset{0 \leq t < |\mathbf{h}_i^c|}{\arg \text{top k}}(\mathbf{S}_i), \quad \mathbf{T}_i \in \mathbb{R}^k \qquad (4)$$

Different $\alpha$ values assign varying weights to local window tokens and yield distinct selection patterns. To further investigate which tokens of the observation window possess a stronger "filter" capability to identify important tokens $\mathbf{T}$, this study explores three methods: 1) **Uniform:** $\alpha = \{1\}_{i=0}^{|\mathbf{h}_i^w|}$. This approach implies that each token in the window produces equivalent contribution when weighted and summed attention scores. 2) **Exponential:** $\alpha = \{2^{i-|\mathbf{h}_i^w|}\}_{i=0}^{|\mathbf{h}_i^w|}$. This approach implies that tokens closer to the end of the window produce higher contribution. 3) **Last Token:** $\text{concat}(\alpha = \{0\}_{i=0}^{|\mathbf{h}_i^w|-1}, \{1\})$. This approach implies that we only consider the last token's attention score in the window.

## 5 EXPERIMENTS

To demonstrate the effectiveness of OmniKV, we conducted extensive experiments on Llama-3-8B-262K, Yi-9B-200K and Llama-3.1-70B-Instruct mainly using the datasets InfiniteBench (Zhang et al., 2024a) and LongBench (Bai et al., 2023). OmniKV demonstrated its effectiveness on 8B and 70B through experiments. Furthermore, we have explored its efficacy on Llama-3.1-405B, with the results presented in Sec D.8 Ablation studies and minor or detailed experiments can be found in Section D.

**Implementation.** For most tasks, we adopted greedy decoding. To prevent repetitive outputs, we employed top-p decoding with $p = 0.95$, temperature $= 0.8$ for summarization tasks in InfiniteBench. All performance and latency experiments were conducted on Nvidia A100 GPUs. Llama-3.1-70B utilized 4-bit weight quantization via bitsandbytes (Dettmers et al., 2021), while other models employed float16 formatting. We makes minor modifications based on Huggingface's transformers (Wolf et al., 2020). For exponential and uniform context selectors, we set the local window size to 16. The "filter" layers $\mathbb{L}$ are set respectively $\{2, 8, 18\}, \{6, 11, 30\}, \{4, 19, 41\}$ for Llama-3-8B-262K, Yi-9B-200K and Llama-3.1-70B-Instruct.

To ensure fairness in comparison, we strictly set OmniKV to retain Mem% of KV caches on GPU. This means we dynamically adjust the token budget for sparse attention layers based on the length of the context prompt length. Mem% consists of two parts: First, OmniKV retains all KV caches for full attention layers, which occupies $\frac{2|\mathbb{L}|+\mathbb{L}_0}{L}$ of the total KV caches. Second, OmniKV's token budget for sparse layers is set to $\frac{|\mathbf{T}|}{|p|}\%$ of KV caches. Therefore, the Mem% of KV cache for OmniKV can be expressed as $\frac{2|\mathbb{L}|+\mathbb{L}_0}{L} + \frac{|\mathbf{T}|}{|p|} \cdot (1 - \frac{2|\mathbb{L}|+\mathbb{L}_0}{L})$. For example, when Mem% is set to 30% and the memory usage of layers $\mathbb{L}$ is 25%, then the token budget is set to 6.7%.

Table 1: Performance of single-step reasoning on LongBench (Bai et al., 2023). *Italics* indicate that the model uses full attention baseline. **Bold** indicates the best performance under the same model. ∼ refers to InfLLM's KV cache memory budget being roughly set to a specific value due to its highly integrated implementation. + indicates dropping tokens within each chunk in H2O due to incompatible with flash attention. Detailed results for every sub-task can be found in Table 4 and 5.

| Methods | %Mem | Single-Doc QA | Multi-Doc QA | Summarize | Few-Shot | Synthetic | Code | Avg. |
|---|---|---|---|---|---|---|---|---|
| *Llama-3-8B-262K* | 100% | 29.2 | 22.9 | 24.9 | 65.9 | 43.5 | 48.9 | 39.2 |
| H2O | 30% | 27.7 | 20.2 | 23.8 | 62.9 | **42.7** | 43.3 | 36.8 |
| InfLLM | ∼30% | 28.1 | 15.3 | 19.1 | 62.7 | 36.5 | 48.5 | 35.0 |
| StreamingLLM | 30% | 19.3 | 17.3 | 18.6 | 49.7 | 11.7 | **52.3** | 28.1 |
| **OmniKV w/ uni** | 30% | **29.6** | **23.3** | 23.7 | 64.0 | 41.5 | 48.4 | 38.4 |
| **OmniKV w/ exp** | 30% | 29.5 | 22.9 | 23.9 | **65.1** | 41.5 | 35.1 | 38.3 |
| **OmniKV w/ last** | 30% | 29.5 | 22.9 | **24.4** | 64.9 | 41.2 | 48.4 | **38.5** |
| *Yi-9B-200K* | 100% | 28.6 | 33.3 | 20.2 | 71.2 | 31.0 | 67.3 | 41.9 |
| H2O | 30% | 26.0 | 33.5 | 17.5 | 68.7 | **31.1** | **66.6** | 40.6 |
| InfLLM | ∼30% | 25.9 | 33.6 | 20.0 | 70.8 | 26.3 | 65.9 | 38.8 |
| StreamingLLM | 30% | 12.4 | 11.3 | 13.5 | 56.3 | 2.9 | 60.7 | 26.2 |
| **OmniKV w/ uni** | 30% | **28.2** | **34.3** | **20.0** | 71.0 | 29.1 | 65.0 | 41.3 |
| **OmniKV w/ exp** | 30% | 28.1 | 34.0 | 20.0 | **71.2** | 30.6 | 65.5 | **41.6** |
| **OmniKV w/ last** | 30% | 28.0 | 33.6 | 19.7 | 70.8 | 30.9 | 65.3 | 41.4 |
| *Llama-3.1-70B* | 100% | 42.2 | 44.8 | 25.5 | 68.9 | 58.0 | 55.7 | 49.2 |
| H2O+ | 20% | 36.6 | 38.9 | 24.1 | 61.2 | 24.5 | 54.0 | 39.9 |
| InfLLM | ∼20% | 39.3 | 36.1 | 18.4 | 62.5 | 41.1 | 39.8 | 39.5 |
| StreamingLLM | 20% | 16.7 | 12.3 | 18.7 | 45.6 | 7.5 | 61.1 | 27.0 |
| **OmniKV w/ uni** | 20% | 40.8 | 43.8 | 23.6 | 67.7 | **57.7** | 53.8 | 47.9 |
| **OmniKV w/ exp** | 20% | 42.0 | 44.3 | 24.6 | **68.4** | 57.6 | **55.2** | **48.7** |
| **OmniKV w/ last** | 20% | **42.0** | **44.3** | **24.7** | 68.2 | 57.6 | 54.8 | 48.6 |

**Baselines.** We employed three state-of-the-art methods for memory reduction as baselines, as well as Full Attention. For methods other than InfLLM and Full Attention, we strictly limited the KV cache size to Mem%. Specific settings and implementations can be found in the Section C. 1) **H2O** (Zhang et al., 2024c), which discards tokens based on attention scores, has been proven to be superior in performance by previous work (Yuan et al., 2024). + indicates dropping tokens within each chunk in H2O due to incompatible with flash attention. 2) **InfLLM** (Xiao et al., 2024a), similar to our method, does not discard any tokens. It divides the sequence into blocks, then chooses a few important tokens in each block as retrieval keys, and retrieves several relevant blocks as context when decoding. 3) **StreamingLLM** (Xiao et al., 2024b) observes that the starting part of the input prompt tokens occupies a large portion of the attention scores, and thus, in addition to the sliding window approach, it retains an initial window as part of the context. 4) **Full Attention** (original model), which does not discard any tokens and uses all tokens as context, serves as the theoretical performance upper bound for comparison.

**Datasets.** To test OmniKV's performance in single-step reasoning, we primarily used two widely applied benchmarks: 1) InfiniteBench (Zhang et al., 2024a) with an average length of 145.1K, covering multiple tasks. We uniformly adopted a 128K context for testing and truncated inputs exceeding 128K at the middle. 2) We tested LongBench's 18 tasks across multiple categories, with most tasks' average length ranging from 5K to 15K (Bai et al., 2023). During testing, all models supported a context length longer than the longest sample, eliminating the need for truncation.

To assess OmniKV's performance in multi-step reasoning, we utilized 2WikiMQA (Ho et al., 2020) and HotpotQA (Yang et al., 2018) from LongBench. However, QA tasks may be biased, as questions often focus on special information, such as a person's birthplace, birth date, or awards received. LLMs might prioritize this type of information during the prefill stage, potentially affecting the accurate evaluation of their long-text capabilities. Motivated by this, we propose a benchmark called **2StageRetr**. This task consists of a dictionary and an equation adding two numbers. LLMs need to use the answer from the equation to search the dictionary, find the corresponding key, and output the value associated with that key. Example and details can be found in Section A.

## 5.1 PERFORMANCE

**Performance with Single-Step Reasoning.** We demonstrate the effectiveness of OmniKV in the Single-Step Reasoning scenarios, which is the standard evaluation method for both benchmarks. In this format, the model receives an input and directly answers the question without providing inter-

Table 2: Performance of single-step reasoning on InfiniteBench (Zhang et al., 2024a).

| Methods | %Mem | En.Sum | En.QA | En.MC | En.Dia | Zh.QA | Code.Debug | Math.Find | Retr.PassKey | Retr.Number | Retr.KV | Avg. |
|---|---|---|---|---|---|---|---|---|---|---|---|---|
| *LLaMA-3-8B-262K* | 100% | 22.0 | 13.3 | 65.9 | 6.0 | 12.8 | 20.8 | 26.5 | 100.0 | 100.0 | 14.4 | 38.1 |
| H2O+ | 30% | 21.8 | **13.8** | 65.5 | **7.0** | 12.5 | 23.0 | **30.5** | 98.4 | 74.0 | 5.2 | 35.2 |
| InfLLM | ∼30% | 17.3 | 9.3 | 44.5 | 4.0 | **17.7** | **29.1** | 24.5 | 100.0 | 100.0 | 0 | 34.6 |
| **OmniKV w/ last** | 30% | **22.5** | 12.7 | **65.5** | 5.0 | 12.6 | 20.0 | 26.5 | **100.0** | **100.0** | 9.6 | **37.4** |
| *Yi-9B-200K* | 100% | 19.3 | 11.1 | 66.3 | 2.0 | 15.6 | 24.1 | 24.8 | 100.0 | 100.0 | 20.2 | 38.3 |
| H2O+ | 30% | **21.6** | 10.8 | 66.3 | 1.0 | **15.8** | 23.3 | 24.0 | 100.0 | 92.7 | 6.0 | 36.1 |
| InfLLM | ∼30% | 6.7 | **12.1** | 37.1 | **3.5** | 14.8 | 21.5 | **34.0** | 100.0 | 100.0 | 0 | 32.9 |
| **OmniKV w/ last** | 30% | 20.0 | 11.1 | **66.8** | 1.0 | 15.5 | **23.8** | 23.8 | **100.0** | **100.0** | 19.8 | **38.2** |

mediate reasoning steps. Results from LongBench and InfiniteBench are presented in Table 1 and 2, respectively. The results indicate that OmniKV achieves the best performance and consistently performs very close to the original full attention baseline across all tasks, showcasing OmniKV's stability. Particularly in Llama-3.1-70B, OmniKV significantly outperforms the baselines. In some task categories, OmniKV even surpasses the baseline results. On InfiniteBench, although OmniKV's performance on the Math.Find task is slightly lower than InfLLM, it remains very close to the original model. However, InfLLM performs poorly on the KV Retrieval task. Thus, the stability of OmniKV's performance, compared to the original model, facilitates its direct application in real-world scenarios without the need for additional testing.

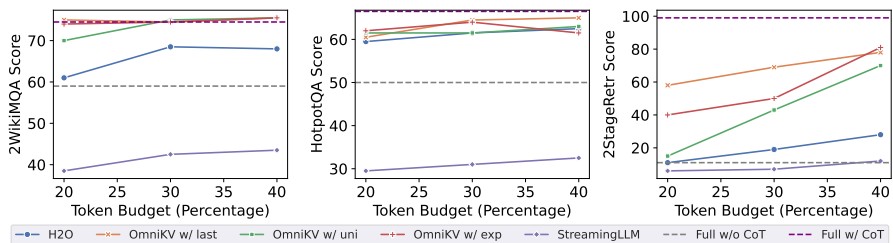

Figure 3: Performance of multi-step reasoning with different token budget ratio of KV cache on three multi-hop tasks.

**Performance with Multi-Step Reasoning.** OmniKV dynamically selects the necessary context for attention computation. To demonstrate the necessity of drop-free and effectiveness of dynamic context selection, we tested models on three multi-step reasoning tasks and adopted the CoT output format to solve the problems. We use **Exact Match** as the metric, meaning that as long as the standard answer appears in the model's output, it is considered correct. The reason for employing a simple metric is due to a more linear metric being better able to quantify performance gaps (Levy et al., 2024; Schaeffer et al., 2023). The results of Llama-3.1-70B are shown in Figure 3. Results show that OmniKV achieved the best performance under all budgets, demonstrating the effectiveness of dynamic context selection. In the 2StageRetr task, we observe that H2O's accuracy cannot exceed its budget, indicating that without prior knowledge, H2O can only randomly retain key-value pairs in task dictionary. We also conducted experiments on the Yi-9B-200K, as detailed in Section D.4.

**Choice of Context Selectors.** In Section 4.2, three context selectors are proposed. From Table 1 and Figure 3, it is evident that in single-step reasoning settings, the exponential and last selectors perform better, while the uniform selector lags slightly but not significantly. In multi-step reasoning settings, the last selector exhibits the best performance, while the uniform selector performs somewhat poorly. The exponential selector can be viewed as an intermediate state between the last and uniform selectors. The fact that the last and exponential selectors outperform the uniform selector suggests that the last may be the most optimal context selector. Moreover, the last method aligns with the pretraining paradigm of LLMs, as the computation of context scores remains entirely consistent with the original model.

From an engineering practice and inference efficiency perspective, last is also the simplest and has the lowest latency, as this selector does not require maintaining cumulative scores for the window. Therefore, in subsequent analysis experiments, we primarily focus on the **last** context selector.

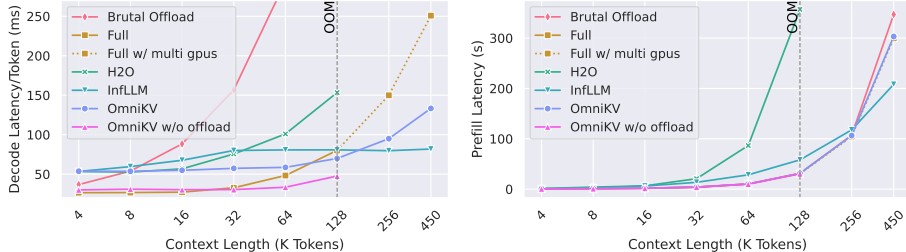

Figure 4: End-to-end latency results. The left and right figures show the latency of the decode and prefill stages respectively. OOM indicates the maximum context length supported by some methods on a single A100 GPU.

## 5.2 LATENCY AND TRADE-OFF

We evaluated the end-to-end inference latency of OmniKV using a single NVIDIA A100 80GB GPU and 12 cores of an Intel Xeon Platinum 8369B CPU at 2.90GHz. A 32-layer LLaMA-3-8B-262K model was utilized, with "filter" layers $\mathbb{L} = \{2, 8, 18\}$, applying flash attention (Dao et al., 2022), and a batch size of 1. For different context length settings, the token budget for sparse attention was set to 2048. During decoding, 50 tokens were generated, and the mean latency per token was calculated over all decoding steps. In the prefill stage, we measured the Time To First Token (TTFT).

The results are presented in Figure 4. The "Brutal Offload" approach refers to offloading all KV cache of the final 20 layers to CPU, and pre-loading all KV cache for each layer 4 layers in advance. InfLLM does not perform full attention during prefill, instead utilizing sparse attention in chunks. Consequently, it achieves better latency at 450K context. However, employing sparse attention in the prefill stage may impact the method's performance (Yuan et al., 2024). OmniKV exhibits identical latency to full attention during the prefill stage, which is attributed to the offload process being covered by the full attention computation. We also tested the original model's latency of 450K context with 3 A100s. Overall, our method demonstrates the best latency performance.

Notably, when GPU memory is sufficient, i.e., within a 128K context for single A100, OmniKV can store the entire KV cache in VRAM. At this point, OmniKV can still perform sparse attention without offloading (OmniKV w/o offload), thus achieving a **1.68x** decoding efficiency of 21.0 tokens/s at 128K context. Moreover, we observe that OmniKV achieves lower latency than Full Attention at 128K. This suggests that we can simply adopt a segmentation strategy for OmniKV to accelerate decoding speed at any context length longer than 32K.

Under settings that save 70% of KV cache, we can run Llama-3-8B with a **450K** context on a single A100 80G GPU at a speed of 7.5 tokens/s. We also validated the performance under 512K ultra-long context on the Passkey task using Llama-3-8B-1048k, achieving perfect results as shown in Figure 7. Under 80% memory reduction, we can run Llama-3.1-70B with a context of **150K** tokens at a speed of 4.5 tokens/s.

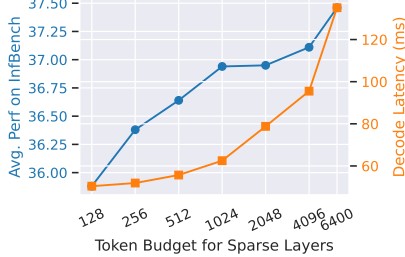

Figure 5: Trade-off on token budgets for sparse layers. Left axis shows the average score on InfiniteBench, and the right axis shows the latency of decoding.

**Trade-off between Performance and Efficiency.** The smaller the token budget used for sparse attention layers, the lower the latency caused by loading KV cache from CPU. We evaluated the trade-off between performance and efficiency on Llama-3-8B-262K for various token budgets on InfiniteBench with 128K context. The results are presented in Figure 5. Here, 6400 represents the token budget under $\mathrm{Mem\%} = 30\%$. This finding indicates that even retrieving only 128 relevant tokens per sparse layer as context can yield satisfactory average score 35.9 higher than H2O+. Furthermore, selecting 1024 tokens achieves a favorable balance between performance and latency. Detailed results could be found in Table 3.

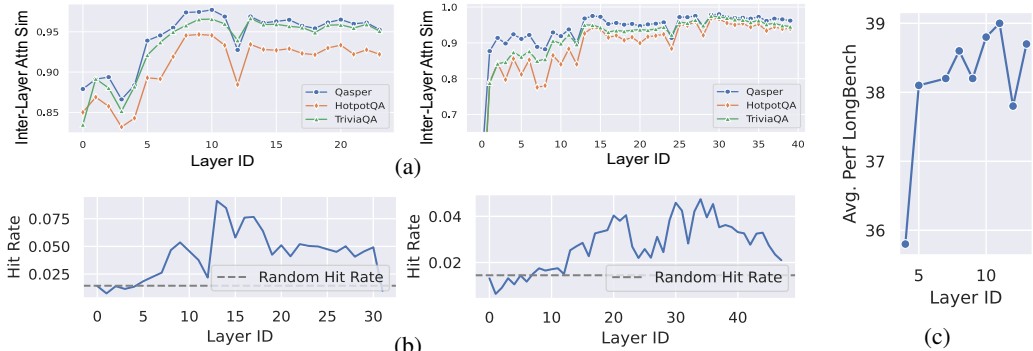

Figure 6: Analysis of "filter" ability in layers. For (a) and (b), left: Llama-3-8B-262K, right: Yi-9B-200K. (a) The inter-layer attention similarity, also referred to as the "filter" capability of layers, displays similar trends across various tasks. (b) Varying capabilities to capture important tokens. (c) Varying performance on LongBench using Llama-3-8B-262K.

## 5.3 ANALYSIS

As shown in Figure 1a, sparse patterns identified by some "filter" layers have higher similarity with subsequent layers than other layers. Naturally, we have the following two research questions: 1) Are these "filter" layers' ability task-dependent or is this more of a model characteristics? 2) Which layers have a greater ability to identify genuinely important tokens? 3) Is the performance varied with "filter" capabilities?

**Task-Independent Filter Layers.** To answer the first research question, we conducted experiments on different tasks to collect those layers' "filter" characteristics. The results, shown in Figure 6a, display the trends of similarity values for Llama-3-8B and Yi-9B across various tasks. We can observe that these curves for different tasks generally follow same trends, suggesting that the strength of "filter" is not task-dependent. Instead, the "filter" ability is more likely an intrinsic characteristic of the layers themselves. This implies that once we select appropriate hyper-parameters $\mathbb{L}$, our method can be adapted to any task.

**Accuracy of Context Selection.** We use the CLongEval dataset (Qiu et al., 2024) to test whether the layers can accurately assign higher attention scores to the context containing the answer. This dataset provides the reference chunk where the standard answer is located. Although this chunk covers a broad range, we can still compute the hit ratio as the proportion of important tokens within the reference chunk. The results are shown in Figure 6b. We observed that the layers with stronger "filter" ability demonstrate a higher hit ratio compared to neighboring layers. For example, the 8th layer in Llama3-8B exhibits a peak in both Figure 6a and 6b. Likewise, the 14th layer in Yi-9B displays similar characteristics. This suggests that certain layers develop a stronger capacity for important token retrieval after training.

**Performance Vary with Filter Ability.** We conduct experiments using Llama3-8B-262K on Long-Bench. As shown in Figure 6c, results indicate that layers 8, 10, 11, and 13 exhibit relatively superior performance, which corresponds well with the higher "filter" ability observed in these layers as shown in Figure 6a. Similarly, the sudden performance decline in layer 12 and the abrupt improvement in performance from layer 4 to layer 5 are also aligned in Figure 6a. Detail results can be found in Section D.6.

## 6 CONCLUSION

This paper proposes OmniKV, a token-dropping-free and training-free inference method, delivering a 1.7x improvement in inference efficiency without compromising performance in long-text scenarios. Moreover, OmniKV is highly compatible with offloading techniques, significantly reducing KV cache memory consumption. The method is simple to implement and has promising practical application prospects. In our future work, we plan to explore the integration of OmniKV with KV cache quantization techniques to further minimize the usage of KV cache and enhance efficiency.

REPRODUCIBILITY STATEMENT

The hyper-parameters, hardware environment, decoding methods, and other pertinent details are presented in Section 5. The core algorithm for OmniKV is provided in Section B. Detailed settings and implementation of baselines can be found in Section C.

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

# A  2STAGERETR

To mitigate the influence of these priors on the results, we propose the benchmark **2StageRetr**. The main idea behind this benchmark is to construct a two-step reasoning problem, using the result of the first step to complete the second step. The task description is placed at the end of the prompt, preventing the model from "mentally calculating" the result of the first step. Thus avoids LLMs assigning high attention scores to answer tokens based on mentally computed results. Specifically, the 2StageRetr consists of a dictionary composed of multiple pairs of numbers and colors, followed by an addition equation. We ensure that the result of the addition is always a key in the dictionary. Then the model needs to find the corresponding color in the dictionary based on the result of the addition, and output it. An example of 2StageRetr is shown below.

> **2StageRetr Example**
>
> Lets play a game. You have a dict and a mathematical addition equation. The keys of a dictionary can be any number. You need to find the corresponding key value in the dictionary after performing the addition and output the value corresponding to that key. The Dict is {0: lime, 1: yellow, 2: red, 3: black, 4: brown, ...... 17: brown, 18: maroon, 19: teal, 20: red, ..... 28: brown, 29: violet}
> The equation is 8 + 10 = ? Answer the corresponding color based on the addition result.
>
> **Example output**: Since 8 + 10 = 18, the corresponding color is "maroon".

This design essentially ensures that LLMs, without prior knowledge of the problem, use pre-trained prior knowledge to assign higher attention scores to certain key-value pairs in the dictionary. Consequently, the H2O method is fundamentally akin to randomly discarding key-value pairs from the dictionary. Hence, its performance cannot surpass the proportion of its token budget. In contrast, our method selects the most relevant context based on the information within the observation window each time, thereby achieving superior performance.

We set the dictionary size to not exceed 200, with numbers arranged sequentially from 0 to the maximum value. This arithmetic sequence should be relatively simple for current LLMs, thereby primarily assessing the model's retrieval capability.

The average length of this dataset is only 739 tokens, with the maximum length being 1382 tokens. However, current cache drop-based methods do not perform very well on this task.

# B  DETAILED IMPLEMENTATION

The actual code implementation required only a minor modification of the code sourced from Huggingface's Transformers library (Wolf et al., 2020). Here we present the core pseudo code of OmniKV in Algorithm 1. This algorithm demonstrate the attention forward procedure of one layer.

# C  BASELINES SETTINGS

Here we provide a detailed description of the settings for different baselines.

1) **H2O**. As H2O requires the output of attention scores, which is incompatible with flash attention, the intermediate activation values directly lead to out of memory errors when processing long sequences. Therefore, when the context length exceeds 60K or when using models larger than 30B, we must calculate attention in chunks and remove tokens based on the attention score within each chunk. We denote this approach with + in our results. To ensure fairness in comparison, we strictly set H2O to retain $\text{Mem}\%$ of KV caches. We also modified code for supporting GQA (Ainslie et al., 2023) in Llama-3 and Yi.

2) **InfLLM**. Since InfLLM uses an LRU-based block cache and has many hyper-parameters, limiting the KV cache by percentage leads to efficiency degradation and performance decline. Therefore, InfLLM is configured to use an average of $\text{Mem}\%$ of KV caches.

---

**Algorithm 1:** Attention Forward of OmniKV

---

**Input:** observation window's hidden states $\mathbf{h}_i^w$, filter layers $\mathbb{L}$, context cache $\mathbf{K}$, $\mathbf{V}$, visible KV cache $\mathbf{K}^v$, $\mathbf{V}^v$, current layer $i$, attention query weight $\mathbf{W}_i^q$, token budget k, window weights $\alpha$

**if** $i \in \mathbb{L}$ **then**

$\quad \mathbf{Q}_i^w \leftarrow \mathbf{W}_i^q \mathbf{h}_i^w$ $\qquad\qquad\qquad\qquad$ ▷ get query states

$\quad \mathbf{A}_i \leftarrow \text{Softmax}\left(\frac{\mathbf{Q}_i^w \mathbf{K}_i^\top}{\sqrt{d}}\right)$ $\qquad\qquad$ ▷ get attention scores

$\quad \mathbf{S}_i \leftarrow \sum_{j=0}^{|\mathbf{h}_i^w|-1} \alpha_j \max_{0 \leq h < H} \mathbf{A}_i[h, j]$ $\qquad$ ▷ get context scores

$\quad \mathbf{T}_i \leftarrow \underset{0 \leq t < |\mathbf{h}_i^c|}{\arg \text{top k}}(\mathbf{S}_i)$ $\qquad\qquad$ ▷ select important tokens

$\quad t \leftarrow$ get the index of $i$ in $\mathbb{L}$

$\quad$ **for** $j = \mathbb{L}_t + 2 \rightarrow \mathbb{L}_{t+1} - 1$ **do**

$\quad\quad | \quad \mathbf{K}_j^v, \mathbf{V}_j^v \leftarrow \text{LoadToGPU}(\mathbf{K}_j[\mathbf{T}_i], \mathbf{V}_j[\mathbf{T}_i])$ $\qquad$ ▷ load subsets of KV cache to GPU

$\quad$ **end**

**end**

**if** $i \in \mathbb{L}$ *or* $i < \mathbb{L}_0$ *or* $i - 1 \in \mathbb{L}$ **then**

$\quad | \quad \mathbf{K}_i^v, \mathbf{V}_i^v \leftarrow \mathbf{K}_i, \mathbf{V}_i$ $\qquad\qquad$ ▷ use original cache for layers performing full attention

**end**

Finally, perform normal Attention with visible KV cache $\mathbf{K}_i^v, \mathbf{V}_i^v$.

---

3) **StreamingLLM**. We strictly set StreamingLLM to retain $\text{Mem}\%$ of KV caches with $1\%$ for initial sink tokens and $\text{Mem} - 1\%$ for local window tokens.

4) **Full Attention** (original model). To avoid excessive GPU memory consumption by intermediate activations in the MLP, we split the hidden states along the sequence length dimension into chunks of size 4000. This reduces peak memory usage without affecting efficiency.

# D  ABLATION STUDIES AND DETAILED EXPERIMENTS

## D.1  NEEDLE-IN-A-HAYSTACK

As OmniKV employs an 8B model capable of supporting a 450K context on a single A100 GPU, we conducted needle-in-a-haystack tests with a maximum context length of 450K, achieving fully accurate performance. We continued to use $\mathbb{L} = \{2, 8, 18\}$, with a token budget of 1024 for sparse layers. We also conducted tests with an input length of 512K on a single NVIDIA H20, also achieving entirely accurate results.The results are shown in Figure 7.

## D.2  DETAILED TRADE-OFF

Here, we present the detailed trade-off between efficiency and performance. Our test results on InfiniteBench are shown in Table 3. The latency here refers end-to-end time of per token in decoding stage.

Table 3: Detailed results on trade-off between performance and efficiency.

| Budget | En.Sum | En.QA | En.MC | En.Dia | Zh.QA | Code.Debug | Math.Find | RT.passkey | RT.Num | RT.KV | Avg. | Latency (ms) |
|---|---|---|---|---|---|---|---|---|---|---|---|---|
| 128 | 19.1 | 12.5 | 64.2 | 4.0 | 10.9 | 20.6 | 26.6 | 100.0 | 100.0 | 1.0 | 35.9 | 50.4 |
| 256 | 19.4 | 13.8 | 62.9 | 6.0 | 11.8 | 20.6 | 26.6 | 100.0 | 100.0 | 2.8 | 36.4 | 51.9 |
| 512 | 20.8 | 12.9 | 63.3 | 4.5 | 11.8 | 20.6 | 26.6 | 100.0 | 100.0 | 5.8 | 36.6 | 55.7 |
| 1024 | 20.5 | 12.9 | 64.6 | 5.5 | 12.1 | 20.6 | 26.6 | 100.0 | 100.0 | 6.6 | 36.9 | 62.5 |
| 2048 | 21.4 | 12.9 | 65.5 | 3.0 | 12.2 | 20.3 | 26.6 | 100.0 | 100.0 | 7.6 | 37.0 | 78.8 |
| 4096 | 20.7 | 12.6 | 65.5 | 4.5 | 12.4 | 20.3 | 26.6 | 100.0 | 100.0 | 8.6 | 37.1 | 95.5 |
| 6400 | 22.5 | 12.7 | 65.5 | 5.0 | 12.6 | 20.0 | 26.5 | 100.0 | 100.0 | 9.6 | 37.4 | 135.1 |

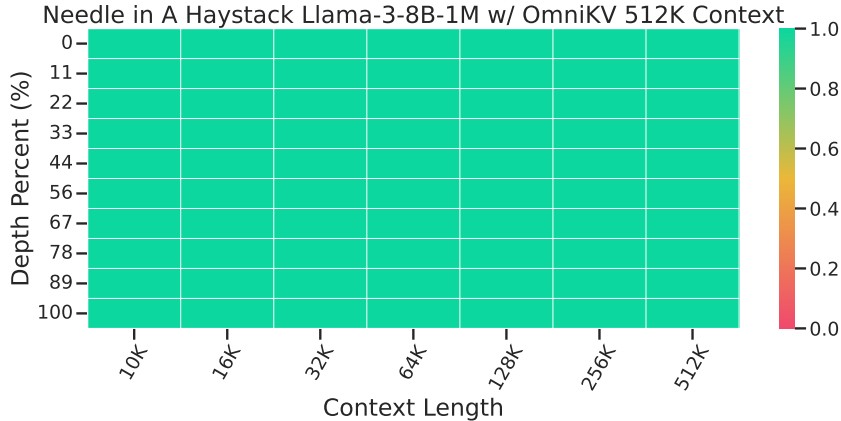

Figure 7: Needle-in-a-haystack test. Results indicate that OmniKV applied to Llama-3-8B-1M achieves perfect retrieval results.

## D.3    DETAILED LONGBENCH RESULTS

We present the detailed results on LongBench, showing the scores for each subtask, as shown in Table 4 and 5.

Table 4: Detailed results on LongBench (Part 1).

| Model | Single-Document QA | | | Multi-Document QA | | | | Summarization | | |
| | MultiFieldQA EN | NarrativeQA | Qasper | 2WikiMQA | Dureader | HotpotQA | Musique | Gov Report | qmsum | vcsum |
|---|---|---|---|---|---|---|---|---|---|---|
| *Llama-3-8B-262K* | 43.6 | 16.0 | 28.2 | 21.7 | 28.3 | 27.5 | 14.3 | 34.5 | 25.3 | 15.1 |
| H2O | 39.8 | 16.8 | 26.5 | 19.3 | 23.9 | 24.5 | 13.2 | 31.2 | 25.3 | 15.2 |
| StreamingLLM | 22.1 | 13.4 | 22.5 | 20.1 | 12.9 | 24.1 | 12.3 | 26.5 | 21.4 | 8.1 |
| InfLLM | 41.2 | 16.8 | 26.2 | 16.5 | 19.8 | 16.5 | 8.4 | 33.1 | 24.2 | 0.2 |
| OmniKV w/ last | 43.0 | 17.0 | 28.6 | 21.8 | 28.2 | 27.0 | 14.6 | 34.2 | 24.9 | 14.1 |
| OmniKV w/ uni | 43.6 | 17.1 | 28.4 | 22.3 | 29.9 | 26.9 | 14.3 | 32.1 | 25.2 | 13.9 |
| OmniKV w/ exp | 43.7 | 16.7 | 28.3 | 21.9 | 28.5 | 26.7 | 14.6 | 33.6 | 24.5 | 13.8 |
| *Yi-9B-200K* | 37.1 | 13.6 | 35.3 | 35.6 | 19.4 | 50.8 | 27.5 | 30.7 | 20.4 | 9.8 |
| H2O | 34.4 | 14.9 | 29.1 | 35.5 | 20.1 | 50.0 | 28.7 | 22.5 | 20.8 | 9.4 |
| StreamingLLM | 17.1 | 2.6 | 17.7 | 19.2 | 10.8 | 10.9 | 4.5 | 19.0 | 15.9 | 5.9 |
| InfLLM | 25.6 | 15.2 | 37.1 | 31.5 | 23.3 | 53.3 | 26.5 | 30.1 | 20.7 | 9.2 |
| OmniKV w/ last | 38.3 | 11.9 | 34.1 | 36.0 | 21.3 | 50.2 | 27.2 | 30.9 | 19.2 | 9.3 |
| OmniKV w/ uni | 38.1 | 12.9 | 33.7 | 35.7 | 21.9 | 51.8 | 27.9 | 31.1 | 20.4 | 8.7 |
| OmniKV w/ exp | 38.7 | 12.7 | 33.1 | 36.0 | 21.7 | 50.6 | 27.8 | 31.0 | 20.1 | 9.0 |
| *Llama-3.1.70B* | 54.9 | 27.9 | 44.1 | 54.5 | 31.2 | 58.8 | 34.8 | 35.2 | 24.0 | 17.5 |
| H2O+ | 45.1 | 29.5 | 35.4 | 42.7 | 27.3 | 52.7 | 32.9 | 32.9 | 23.0 | 16.6 |
| StreamingLLM | 25.3 | 10.1 | 14.9 | 19.5 | 10.0 | 15.5 | 4.4 | 26.7 | 19.8 | 9.8 |
| InfLLM | 50.2 | 24.7 | 43.2 | 47.3 | 32.7 | 43.2 | 21.4 | 19.9 | 20.1 | 15.3 |
| OmniKV w/ last | 54.8 | 27.5 | 43.7 | 56.2 | 28.0 | 59.3 | 33.9 | 33.5 | 24.2 | 16.5 |
| OmniKV w/ uni | 53.7 | 26.5 | 42.3 | 55.9 | 26.2 | 58.6 | 34.6 | 31.6 | 23.9 | 15.5 |
| OmniKV w/ exp | 53.8 | 27.7 | 44.5 | 56.0 | 28.3 | 57.8 | 35.3 | 33.6 | 24.2 | 16.2 |

## D.4    MULTI-STEP REASONING RESULTS OF YI-9B-200K

To further validate the effectiveness of OmniKV in multi-step reasoning, we continued experiments using Yi-9B-200K. The average length of 2StageRetr is 739, so the token budget is $739 \times 0.067 = 49$. To avoid an excessively low budget, while ensuring fairness in comparison, we set the "filter" layers $\mathbb{L} = \{3, 11, 30\}$ to allocate more token budgets for sparse layers.

As shown in Table 6, OmniKV achieved the best results across all three datasets, particularly under a constrained token budget. This further validates the effectiveness of our approach. Yi-9B-200K does not follow instruction to directly answer in 2StageRetr.

## D.5    ABLATION STUDIES

OmniKV comprises a Context Selector and a Context Bank, which are highly coupled modules. When the Context Selector is removed, implying the computation of entire context, the method

Table 5: Detailed results on LongBench (Part 2).

| Model | Few-Shot | | | Syntheic Tasks | | Code | |
|---|---|---|---|---|---|---|---|
| | lsht | trec | TriviaQA | Passage Count | Passage Retrieval EN | lcc | RepoBench-p |
| *Llama-3-8B-262K* | 44.5 | 69.5 | 83.7 | 0.0 | 87.0 | 52.5 | 45.4 |
| H2O | 35.0 | 68.5 | 85.5 | 0.0 | 85.5 | 42.8 | 43.8 |
| StreamingLLM | 20.5 | 60.0 | 68.7 | 1.7 | 21.8 | 55.8 | 48.8 |
| InfLLM | 37.0 | 68.5 | 82.7 | 6.5 | 66.5 | 52.3 | 44.8 |
| OmniKV w/ last | 44.5 | 69.5 | 80.9 | 0.0 | 82.5 | 51.4 | 45.4 |
| OmniKV w/ uni | 41.5 | 68.5 | 82.0 | 0.0 | 83.0 | 51.4 | 45.4 |
| OmniKV w/ exp | 43.5 | 69.5 | 82.5 | 0.0 | 83.0 | 37.5 | 32.9 |
| *Yi-9B-200K* | 48.5 | 78.5 | 86.8 | 3.0 | 59.0 | 71.6 | 63.1 |
| H2O | 42.0 | 77.5 | 86.8 | 2.8 | 59.5 | 70.7 | 62.6 |
| StreamingLLM | 24.3 | 69.5 | 75.2 | 3.1 | 2.8 | 64.4 | 57.1 |
| InfLLM | 48.0 | 77.0 | 87.6 | 3.2 | 49.5 | 69.4 | 62.4 |
| OmniKV w/ last | 48.5 | 78.5 | 85.5 | 3.3 | 58.5 | 69.5 | 61.3 |
| OmniKV w/ uni | 48.0 | 78.5 | 86.7 | 2.8 | 55.5 | 68.9 | 61.1 |
| OmniKV w/ exp | 48.5 | 78.5 | 86.6 | 3.3 | 58.0 | 69.7 | 61.3 |
| *Llama-3.1.70B* | 46.0 | 75.0 | 85.9 | 18.5 | 97.5 | 48.2 | 63.3 |
| H2O+ | 30.0 | 68.5 | 85.1 | 9.5 | 39.5 | 50.4 | 57.6 |
| StreamingLLM | 4.0 | 54.5 | 78.4 | 1.5 | 13.6 | 67.3 | 55.0 |
| InfLLM | 36.0 | 65.5 | 86.0 | 5.5 | 76.8 | 32.3 | 47.4 |
| OmniKV w/ last | 44.5 | 74.3 | 85.9 | 18.0 | 97.3 | 49.2 | 60.5 |
| OmniKV w/ uni | 42.0 | 74.5 | 86.8 | 18.0 | 97.5 | 47.8 | 59.8 |
| OmniKV w/ exp | 44.0 | 75.0 | 86.4 | 18.0 | 97.3 | 48.7 | 61.8 |

Table 6: Multi-step reasoning results of Yi-9B-200K.

| Model | CoT | %Mem | 2WikiMQA | HotpotQA | 2StageRetr |
|---|---|---|---|---|---|
| *Yi-9B-200K* | ✗ | 100.0% | 27.0 | 40.0 | - |
| *Yi-9B-200K* | ✓ | 100.0% | 64.0 | 48.5 | 36.0 |
| H2O | ✓ | 30.0% | 55.5 | 45.5 | 13.0 |
| H2O | ✓ | 40.0% | 60.5 | 47.0 | 19.0 |
| H2O | ✓ | 50.0% | 64.0 | 50.0 | 26.0 |
| OmniKV | ✓ | 30.0% | 58.0 | 51.0 | 14.0 |
| OmniKV | ✓ | 40.0% | 62.0 | 51.0 | 32.0 |
| OmniKV | ✓ | 50.0% | 64.0 | 50.0 | 31.0 |

becomes equivalent to full attention if GPU memory is sufficient. Otherwise, we have to offload some layers' KV cache, then the approach becomes identical to Brutal Offload. Although the performance of Brutal Offload is entirely equivalent to that of the original model, the frequent loading and offloading during the decode stage incurs significant overhead.

Without the Context Bank, indicating that we do not apply inter-layer attention similarity, we can only utilize the Context Selector at each layer to choose the token set **T**. However, selecting important tokens/context itself requires full attention at the current layer, which means efficiency would be significantly reduced. The only potential benefit might be using a shorter context to avoid interference from irrelevant information, possibly leading to better performance.

Table 7: Detailed results on trade-off between performance and efficiency.

| Variant | En.Sum | En.QA | En.MC | En.Dia | Zh.QA | Code.Debug | Math.Find | RT.passkey | RT.Num | RT.KV | Avg. | Latency (ms) |
|---|---|---|---|---|---|---|---|---|---|---|---|---|
| OmniKV w/o CS | 22.0 | 13.3 | **65.9** | **6.0** | 12.8 | 20.8 | 26.5 | 100.0 | 100.0 | **14.4** | **38.1** | 569.4 |
| OmniKV w/o CB | 21.8 | **13.5** | 61.5 | 4.5 | **13.6** | **21.5** | 26.2 | 100.0 | 100.0 | 9.8 | 37.2 | 184.4 |
| **OmniKV** | **22.5** | 12.7 | 65.5 | 5.0 | 12.6 | 20.0 | **26.5** | **100.0** | **100.0** | 9.6 | 37.4 | **54.3** |

## D.6 PERFORMANCE OF FILTER LAYERS

Theoretically, if we select "filter" layers with stronger "filter" capabilities, we can expect improved performance. However, ensuring a completely fair performance comparison when testing different

selected layers is challenging. This is due to the sequential nature of the layers; even if we ensure that the total number of layers performing full attention remains consistent, there can still be issues with uneven spacing between "filter" layers. Attempting to evenly distribute the "filter" layer set introduces additional variables. Nonetheless, experimental results indicate that layers with enhanced "filter" capabilities tend to exhibit superior performance to some extent. We conduct experiments on both LongBench and InfiniteBench. The results are shown in Table 8.

Table 8: Performance on LongBench of different filter layers settings.

| Filter Layers | Single-Doc QA | Multi-Doc QA | Summarization | Few-Shot | Synthetic | Code | Avg. |
|---|---|---|---|---|---|---|---|
| 2,4,18 | 27.0 | 19.9 | 21.4 | 58.1 | 42.8 | 46.0 | 35.8 |
| 2,5,18 | 29.3 | 22.8 | 23.4 | 63.4 | 42.5 | 47.1 | 38.1 |
| 2,7,18 | 29.3 | 22.1 | 23.6 | 65.2 | 41.8 | 47.2 | 38.2 |
| 2,8,18 | 29.6 | 22.9 | **24.4** | 65.0 | 41.3 | **48.4** | 38.6 |
| 2,9,18 | 28.6 | 22.5 | 24.1 | 65.1 | 41.8 | 47.4 | 38.2 |
| 2,10,18 | 30.2 | **23.0** | 24.0 | 65.1 | 42.8 | 47.5 | 38.8 |
| 2,11,18 | **31.0** | 22.7 | 23.9 | **65.4** | **43.0** | 48.2 | **39.0** |
| 2,12,18 | 29.8 | 21.2 | 23.0 | 63.2 | 42.1 | 47.8 | 37.8 |
| 2,13,18 | 30.1 | 22.5 | 23.7 | 65.4 | 42.5 | 48.2 | 38.7 |

### D.7 AN EXAMPLE OF INTER-LAYER ATTENTION SIMILARITY MAP

To directly observe the "filter" capability of layers, we also demonstrated the similarity between any two layers. We visualized the cumulative attention scores for the top-2048 token set from Llama-3-8B-262K on HotpotQA as a measure of similarity, consistently as before. As shown in Figure 8, for layer 8, despite being 12 layers apart from layer 20, the token set $T_8$ selected by layer 8 still achieves a cumulative attention score of **0.87** at layer 20. This substantially demonstrates the effective ability of layer 8 to select important tokens.

### D.8 EFFECTIVENESS OF OMNIKV ON LARGER MODELS

We first visualized the Inter-Layer Attention Similarity of Llama-3.1 405B, as illustrated in Figure 9. The 405B model continues to exhibit remarkably high inter-layer similarity, indicating that OmniKV can be effectively applied to it. Additionally, we conducted further evaluations on two tasks from LongBench, Qasper and Qmsum. And the results are presented in Table 9.

Table 9: OmniKV in 405B.

| Setting | qasper | qmsum |
|---|---|---|
| Llama 3.1 405B | 50.0 | 25.5 |
| OmniKV | 48.5 | 25.9 |

### D.9 COMPATIBILITY OF OMNIKV

In this paper, most of the experiments are conducted using the Huggingface Transformers. However, this framework is typically not used as an inference engine. Current engines used for large model inference, such as vLLM (Kwon et al., 2023a) and LightLLM (ModelTC, 2023), are much faster than those based on Hugging Face Transformers. Therefore, we adapted OmniKV to one of them, LightLLM, to demonstrate that OmniKV has a excellent usability in real-world scenarios. The results are shown in Table 10.

**Continuous Batching.** Continuous Batching is a technique used in the training and inference of Large Language Models (LLMs) to optimize computational efficiency, particularly in scenarios where the model processes sequences of varying lengths. The core idea behind continuous batching is to dynamically group input sequences into batches that can be processed in parallel, even if the sequences have different lengths. OmniKV can be natually integrated with Continuous Bathcing. Based on lightLLM, we find that the continuous batching technique decrease the average latency of multiple 64K-length requests from 41.7s to 37.8s.

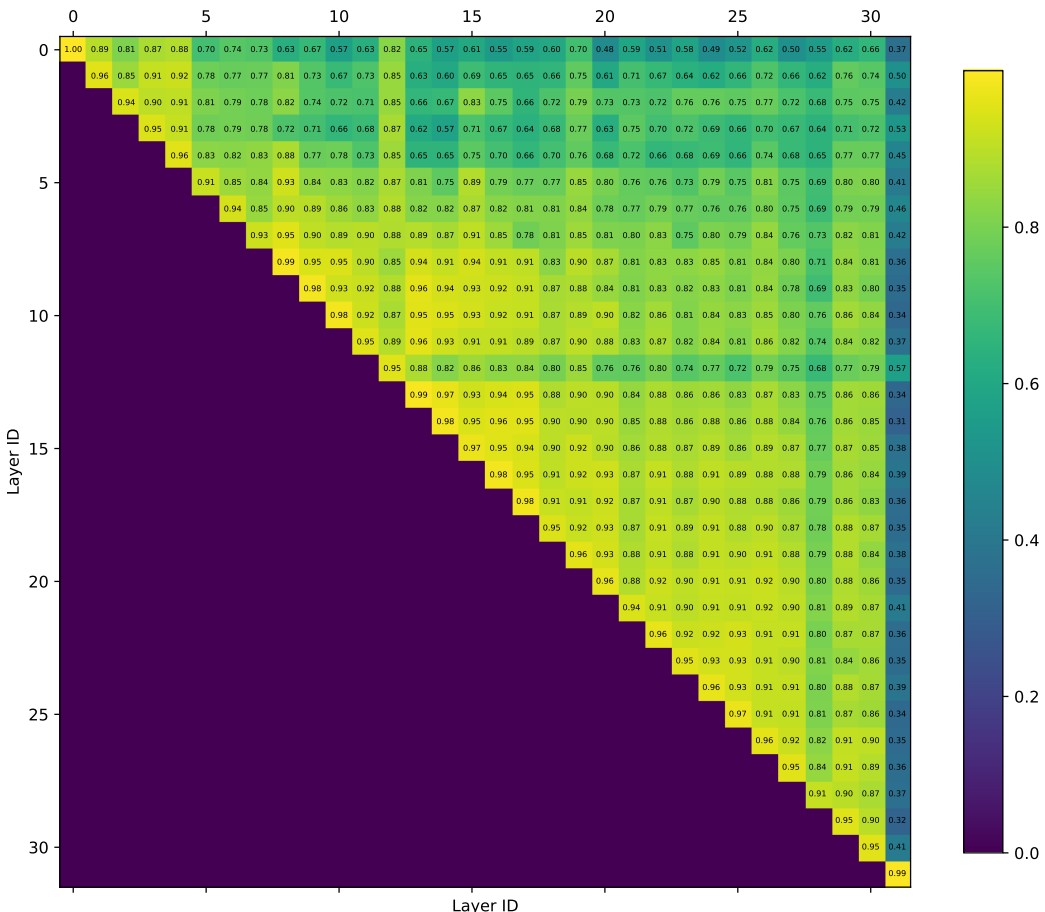

Figure 8: An Example of Inter-Layer Attention Similarity Map.

Table 10: Average latency of per token for each request (ms/token, tp=4).

| Setting | Lightllm+OmniKV | Lightllm+origin | vLLM+origin |
|---|---|---|---|
| 128k, bs=16 | **46.2** | 73.5 | 72.3 |
| 256k, bs=8 | **44.9** | 75.4 | 73.1 |
| 512k, bs=4 | **44.9** | 78.1 | 75.6 |

**Tensor parallelism.** Tensor parallelism(TP) is a technique used in large language models (LLMs) and other deep learning models to distribute the computation of tensors (multi-dimensional arrays) across multiple devices, such as GPUs. This approach is particularly useful for training and inference with very large models that cannot fit into the memory of a single device. Using LightLLM, we have implemented OmniKV with tensor parallelism. Experimental shown in Table 11 results indicate that with TP enabled, OmniKV can still achieve considerable acceleration compared the original model. Another observation is that Lightllm+OmniKV experiences reduced decoding latency as sequence length increases. This is because the "non-filter layers" have a fixed sequence length of 2048. As the batch size decreases, the computation also decreases for "non-filter layers".

**More technical details of TP.** The attention heads is distributed across different GPUs in tensor parallelism, and the outputs are aggregated using all-reduce, making inter-GPU communication a key factor in efficiency. For tp $= n$ , OmniKV maintains $n$ indices of important tokens for each card. Each index of important tokens is computed solely by the card's own attention heads, thereby eliminating the need for inter-GPU communication. Since different attention heads may

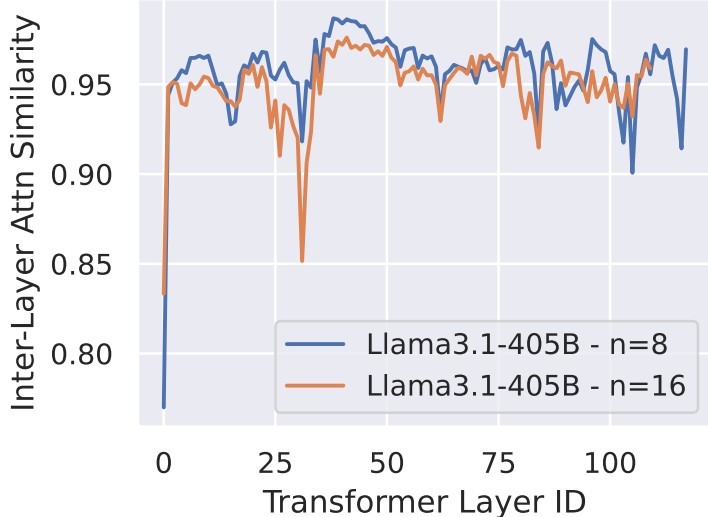

Figure 9: Inter-Layer Attention Similarity of Llama-3.1 405B.

Table 11: Throughput results of decoding (tp=4), using Llama-3-8B-Instruct-262K.

| Setting | Lightllm+OmniKV | Lightllm+origin | vLLM+origin |
|---------|-----------------|-----------------|-------------|
| 128k, bs=16 | **345.9** | 217.7 | 221.3 |
| 256k, bs=8 | **177.5** | 106.1 | 109.4 |
| 512k, bs=4 | **89.1** | 51.2 | 52.9 |

have distinct indices, the system intuitively exhibits greater flexibility. We have also substantiated this through experiments(Shown in Table 12).

Table 12: Performance comparison across datasets.

| Setting | 2WikiMQA | Qasper | HotpotQA | lcc | qmsum |
|---------|----------|--------|----------|-----|-------|
| tp=8 | 21.7 | 28.0 | **27.3** | **52.3** | **25.9** |
| tp=1 | **21.8** | **28.6** | 27.0 | 51.4 | 24.9 |

**Pipeline parallelism**. Pipeline parallelism(PP) is a technique used to parallelize the execution of LLMs across multiple devices or processors. This approach is particularly useful for deep neural networks, where the model can be divided into smaller, sequential stages, and each stage can be processed on a different device. OmniKV is also natually compatible with pipeline parallelism. The experiments of Llama-3.1-70B shown in Table 1 is conducted using pipeline parallelism.

**Context parallelism**. Context parallelism(CP) is a technique used in large language models (LLMs) to improve training and inference efficiency by distributing the processing of input sequences across multiple devices or processors. This approach leverages the fact that many LLMs, such as transformers, process input sequences in a way that allows for parallel computation. Indeed, when OmniKV is used, there is no need to enable context parallelism. This is due to OmniKV only needs around 2048 tokens to achieve excellent results. Nevertheless, OmniKV can still be integrated with context parallelism. In this case, the communication latency in multi-GPU parallelism often exceeds the computational latency of a single GPU. Hence, we can employ context parallelism only in dense (full) attention layers, while utilizing a single GPU for sparse attention layers. This means we simply need to configure which layers will implement context parallelism. We can allocate samples to different GPUs when the batch size is larger than 1 to prevent resource wastage.

## D.10 PREFILL ACCELERATION

OmniKV is designed to enhance **decoding** speed. However, given the importance of prefill acceleration, we have modified OmniKV to optimize the prefill stage as well. Experimental results indicate that OmniKV-prefill achieves a **1.90**x reduction in latency without performance loss compared to the original model with an input length of 256K. The code here is implemented using the Transformers library, enhanced with Flash Attention.

Specifically, during the prefill stage, we divide the sequence dimension into multiple chunks for computation. For each chunk, we also use the "filter layer" to select important tokens. Since the sequence length of the query chunk is no longer 1, we employ the uniform computation method introduced in Section 4.2 to select important tokens in the "filter layer".

Table 13 and Table 14 show latency and quality preserving results. Here, "r" denotes the size of the selected tokens, with the ratio of token size to chunk size being "r". Thus, the larger the value of r, the less information is lost. The results indicate that r=2 has achieved a fine balance between efficiency and performance.

Table 13: Prefill latency (s) on a single NVIDIA-H20.

| Setting | 256k | 128k | 64k | 32k | 16k |
|---|---|---|---|---|---|
| OmniKV-prefill, r=1 | 105.9 | 36.7 | 14.3 | 6.2 | 2.8 |
| **OmniKV-prefill, r=2** | **107.2** | **38.8** | **15.9** | **6.5** | **2.9** |
| OmniKV-prefill, r=4 | 112.9 | 42.8 | 17.1 | 6.8 | 3.0 |
| OmniKV-prefill, r=8 | 133.3 | 49.2 | 19.0 | 6.9 | 2.9 |
| MInference | 61.6 | 28.1 | 13.0 | 6.5 | 3.7 |
| Llama-3-8B-262K (origin) | 203.8 | 58.6 | 18.3 | 6.5 | 2.6 |

MInference (Jiang et al., 2024a) is specifically designed for **Prefill** acceleration and optimizes multiple CUDA kernels. But OmniKV solely relies on the Huggingface-transformers library, consequently less efficient.

Table 14: Quality preserving results on LongBench.

| Setting | %Mem | Single-Doc QA | Multi-Doc QA | Summarization | Few-Shot | Synthetic Tasks | Code |
|---|---|---|---|---|---|---|---|
| Llama-3-8B-262K (origin) | 100% | 29.2 | 22.9 | 24.9 | 65.9 | 43.5 | 48.9 |
| Minference | 30% | 29.2 | 22.4 | **25.3** | 65.8 | 41.0 | 49.8 |
| OmniKV-prefill, r=1 | 30% | 29.8 | 22.0 | 24.0 | 63.5 | 33.3 | **50.0** |
| **OmniKV-prefill, r=2** | 30% | **30.0** | 23.1 | 24.3 | 65.1 | 40.0 | 49.6 |
| OmniKV-prefill, r=4 | 30% | 29.8 | 22.9 | 24.3 | 65.9 | 42.0 | 49.1 |
| OmniKV-prefill, r=8 | 30% | 29.8 | 22.9 | 24.3 | **65.9** | 42.0 | 49.0 |
| OmniKV-prefill, all | 30% | 29.6 | **23.3** | 23.7 | 64.0 | 41.5 | 48.4 |

