# OpenReview forum: "OmniKV: Dynamic Context Selection for Efficient Long-Context LLMs"
_ICLR.cc/2025/Conference — ICLR 2025 Poster_

### Official Review · Reviewer_UJXu · 2024-11-01

**Soundness:** 2
**Presentation:** 3
**Contribution:** 2
**Rating:** 6
**Confidence:** 4

**Summary:**

The paper presents OmniKV to reduce GPU memory usage when serving Large Language Models (LLMs) with long contexts. OmniKV is based on the observation that the importance of history tokens cannot be determined by the current stage attention score. Thus, permanently discarding some of past KV tokens may incur accuracy loss. OmniKV is a token-dropping-free and training-free inference method that reduces KV cache GPU memory usage by over 75% without performance degradation, which also accelerates  inference efficiency in long-text scenarios. The key insight of OmniKV is that important tokens are very similar across consecutive LLM layers. So that the KV Cache, which is offloaded to CPU, can be loaded to GPU in the meantime for different layers. Results demonstrate that OmniKV can handle a 450K context with a decoding latency of 7.52 tokens/s, which is 1.87 times faster than the original model running on three A100 GPUs in pipeline mode.

**Strengths:**

1) This paper focuses on a timely and important problem: Reducing the KV Cache memory usage of long-context LLMs.

2) OminKV greatly reduces the GPU memory consumption when serving long-context LLM through offloading. It reduces the LLM inference latency by only loading important history tokens to GPU for attention computation. OmniKV well preserves model’s performance on long-context benchmarks such as LongBench and InfiniteBench.

3) OmniKV notices that the important tokens for consecutive layers are similar, merging the CPU-GPU memory loading procedure for multiple layers to reduce memory transaction and important token identification overheads.

**Weaknesses:**

1) The optimization techniques proposed in this paper cannot accelerate the prefilling speed of LLM.

2) In Figure 4, it seems that OmniKV (w/ offload) is slower than the dense baseline until the sequence length reaches 128K. For OmniKV w/o offload, the speed also lags behind the dense baseline when sequence length < 32K.

3) The dense baseline (Full) seems not to be strong enough. For Llama-3-8B model on A100, TensorRT-LLM and vLLM take less than 20ms to decode a token (sequence length = 128K, batch size = 1) . However, the dense baseline takes ~80ms,  according to Figure 4.

**Questions:**

1) What is the dense baseline implementation used for efficiency evaluation in Figure 4?
2) In the last sentence of abstract, the authors mention that “By using a single A100 and Llama-3-8B, OmniKV can handle a 450K context with a decoding latency of 7.52 tokens/s, which is 1.87 times faster than the original model running on three A100 GPUs in pipeline.” I would appreciate if the authors could elaborate i) what is the baseline setting for “original model running on three A100 GPUs in pipeline”; ii) what will be the latency for “original model” if the model is served with tensor parallelism rather than pipeline parallelism.

---

> ### Author Response · Authors · 2024-11-21
> **Response (1/2) to Reviewer UJXu**
>
> > The optimization techniques proposed in this paper cannot accelerate the prefilling speed of LLM.
>
> We have modified OmniKV to optimize the prefill stage as well. Experimental results indicate that OmniKV-prefill achieves a **1.90**x reduction in latency without performance loss compared to the original model with an input length of 256K.
>
> Specifically, during the prefill stage, we divide the sequence dimension into multiple chunks for computation. For each chunk, we also use the "filter layer" to select important tokens. Since the sequence length of the query chunk is no longer 1, we employ the uniform context selector introduced in Section 4.2 to select important tokens in the "filter layer".
>
> Here, "r" denotes the size of the selected tokens, with the ratio of token size to chunk size being "r". Thus, the larger the value of r, the less information is lost.
>
> The results indicate that r=2 has achieved a fine balance between efficiency and performance.
>
> **Prefill latency (s) on a single NVIDIA-H20:**
>
> | Exp. settings            | 256k  | 128k | 64k  | 32k | 16k |
> | ------------------------ | ----- | ---- | ---- | --- | --- |
> | OmniKV-prefill, r=1      | 105.9 | 36.7 | 14.3 | 6.2 | 2.8 |
> | **OmniKV-prefill, r=2**  | 107.2 | 38.8 | 15.9 | 6.5 | 2.9 |
> | OmniKV-prefill, r=4      | 112.9 | 42.8 | 17.1 | 6.8 | 3.0 |
> | OmniKV-prefill, r=8      | 133.3 | 49.2 | 19.0 | 6.9 | 2.9 |
> | MInference               | 61.6  | 28.1 | 13.0 | 6.5 | 3.7 |
> | Llama-3-8B-262K (origin) | 203.8 | 58.6 | 18.3 | 6.5 | 2.6 |
>
> **Quality preserving results on LongBench:**
>
> | Exp. settings            | %Mem | Single-Doc QA | Multi-Doc QA | Summarization | Few-Shot | Syntheic Tasks | Code     |
> | ------------------------ | ---- | ------------- | ------------ | ------------- | -------- | -------------- | -------- |
> | Llama-3-8B-262K (origin) | 100% | 29.2          | 22.9         | 24.9          | 65.9     | 43.5           | 48.9     |
> | Minference               | 30%  | 29.2          | 22.4         | **25.3**      | 65.8     | 41.0           | 49.8     |
> | OmniKV-prefill, r=1      | 30%  | 29.8          | 22.0         | 24.0          | 63.5     | 33.3           | **50.0** |
> | **OmniKV-prefill, r=2**  | 30%  | **30.0**      | 23.1         | 24.3          | 65.1     | 40.0           | 49.6     |
> | OmniKV-prefill, r=4      | 30%  | 29.8          | 22.9         | 24.3          | 65.9     | 42.0           | 49.1     |
> | OmniKV-prefill, r=8      | 30%  | 29.8          | 22.9         | 24.3          | **65.9** | 42.0           | 49.0     |
> | OmniKV-prefill, all      | 30%  | 29.6          | **23.3**     | 23.7          | 64.0     | 41.5           | 48.4     |
>
> > In Figure 4, it seems that OmniKV (w/ offload) is slower than the dense baseline until the sequence length reaches 128K. For OmniKV w/o offload, the speed also lags behind the dense baseline when sequence length < 32K.
>
> Since we have integrated OmniKV with [Lightllm](https://github.com/ModelTC/lightllm), we have further reduced decoding latency. Our experiments demonstrate that decoding acceleration can be achieved when the sequence length is ≥16k. It is important to note that the half-precision computational power of the H20 and A100 is **148** and **312** TFLOPS, respectively.
>
> **Decoding latency (ms/token) on a single NVIDIA-H20:**
>
> | Exp. settings | Lightllm+OmniKV | Lightllm+origin |
> |:------------- | --------------- | --------------- |
> | 16K           | **18.5**        | 19.6            |
> | 32K           | **18.7**        | 21.4            |
> | 64K           | **18.9**        | 24.9            |
> | 128K          | **22.1**        | 32.3            |
> | 256K          | **26.4**        | 47.1            |
>
> Moreover, these overheads can be entirely avoided by dynamically adjusting computation method based on input length. For instance, when the input length is 8K, the original model can be utilized, whereas for a length of 32K, OmniKV can be employed.

---

> ### Author Response · Authors · 2024-11-21
> **Response (2/2) to Reviewer UJXu**
>
> > The dense baseline (Full) seems not to be strong enough. For Llama-3-8B model on A100, TensorRT-LLM and vLLM take less than 20ms to decode a token (sequence length = 128K, batch size = 1) . However, the dense baseline takes ~80ms, according to Figure 4.
> >
> > What is the dense baseline implementation used for efficiency evaluation in Figure 4? i) what is the baseline setting for “original model running on three A100 GPUs in pipeline”
>
> The Full (dense) Attention baseline we use here is based on Huggingface's Transformers library, employing FlashAttention2 as the backend. However, this implementation lacks kernel-level optimization, resulting in significantly slower decoding speeds compared to inference frameworks like vLLM.
>
> As we implemented OmniKV on Lightllm, and as shown in the previous table, our method still demonstrated better latency. Because the half-precision computational power of the NVIDIA-H20 and A100 is **148** and **312** TFLOPS, the latency on NVIDIA-H20 (seq len=128k, bs=1) is higher than A100.
>
>
>
> > ii) what will be the latency for “original model” if the model is served with tensor parallelism rather than pipeline parallelism.
>
> We have made OmniKV compatible with tensor parallelism, and OmniKV achieves superior results compared to the original model on vLLM with tensor parallelism.
>
> **Average latency of per token for each request (ms/token, tp=4):**
>
> | Exp. settings | Lightllm+OmniKV | Lightllm+origin | vLLM+origin |
> | ------------- | --------------- | --------------- | ----------- |
> | 16k, bs=100   | **68.9**        | 70.5            | 71.6        |
> | 32k, bs=64    | **58.0**        | 79.6            | 79.8        |
> | 64k, bs=32    | **51.0**        | 77.2            | 72.6        |
> | 128k, bs=16   | **46.2**        | 73.5            | 72.3        |
> | 256k, bs=8    | **44.9**        | 75.4            | 73.1        |
> | 512k, bs=4    | **44.9**        | 78.1            | 75.6        |
>
> Lightllm+OmniKV demostrates reduced decoding latency as sequence length increases. This is because the "non-filter layers" have a fixed sequence length of 2048. As the batch size decreases, the computation also decreases for "non-filter layers". But when context length decreases, the computation remains the same for "non-filter layers".

---

> > ### Comment · Reviewer_UJXu · 2024-11-24
> >
> > I would like to thank the authors for the detailed response. And I still have a few follow-up concerns regarding the clarifications provided:
> >
> > Regarding W1: Could you further elaborate how do you combine OmniKV with chunk-prefilling? -- The authors mention that "r denotes the size of the selected tokens, with the ratio of token size to chunk size being r". I interpret this to mean that "r" represents the number of chunks into which the context has been divided. Therefore, I am wondering what is the difference between "r=1" and the baseline? The paper mentions that OmniKV “perform full attention" in "all layers" in prefilling stage, why is it faster than the baseline? Do you evict tokens in the prefilling stage in the current version?

---

> > > ### Author Response · Authors · 2024-11-29
> > >
> > > Dear Reviewer UJXu,
> > >
> > > Thank you sincerely for your invaluable and constructive feedback. We deeply appreciate your thoughtful insights and have made every effort to address each of your concerns comprehensively.
> > >
> > > We would be grateful if you could let us know if our responses have satisfactorily addressed your concerns. We also would like to know if our revisions meet the criteria for an increase in score. As always, we welcome any additional questions or suggestions you may have, as we are committed to ongoing discussion.
> > >
> > > Best regards, The Authors

---

> > > > ### Comment · Reviewer_UJXu · 2024-11-29
> > > >
> > > > Thank you very much for the clarification. Now I understand the details of the authors' modification on OmniKV (optimize the prefill stage). The new integration to lightllm makes the efficiency of OmniKV much better than the results reported in the original paper. Could you please clarify in the table of **"Decoding latency (ms/token) on a single NVIDIA-H20"**, did you activate offloading for OmniKV? Thanks.

---

> > > > > ### Author Response · Authors · 2024-12-01
> > > > >
> > > > > Dear Reviewer UJXu,
> > > > >
> > > > > Thank you for your valuable feedback. As the discussion phase is nearing completion, we would like to know whether our responses have addressed your concerns. We are also happy to answer any additional questions you might have.
> > > > >
> > > > > We appreciate your insightful review and look forward to your feedback.
> > > > >
> > > > > Best regards, The Authors

---

> > > > > > ### Comment · Reviewer_UJXu · 2024-12-01
> > > > > >
> > > > > > Thank you for the previous clarification.
> > > > > >
> > > > > > I would like to know do you have any quantitative estimation on the overhead introduced by offloading with the current implementation? As the Lightllm is a much more efficient framework, I am concerned that the offloading could significantly impact the end-to-end latency.
> > > > > >
> > > > > > Offloading is a very important feature for OmniKV. OmniKV is "token-dropping-free". Offloading can help to address the KV Cache memory issue and has been widely discussed in the original paper. Therefore, I would like to know is there any way to reproduce/simulate the results for OmniKV w/ offloading as in Figure 4?

---

> ### Author Response · Authors · 2024-11-25
> **Response to Reviewer UJXu**
>
> Thank you for your thoughtful follow-up questions and for taking the time to review our responses in such detail. We appreciate the opportunity to further clarify and address your concerns.
>
> > I interpret this to mean that "r" represents the number of chunks into which the context has been divided.
>
> There might be some misunderstanding here. The parameter $r$ determines the **size of selected important tokens**, which is calculated as $r \times$ chunk\_size. Here, chunk\_size is a hyperparameter that represents the size of a single chunk. Please note that OmniKV-prefill does not select $r$ chunks, but $r \times $ chunk\_size tokens. Token level selection offers better flexibility.
>
> Specifically, as described in Section 4.2, OmniKV can use a local observation window to identify important tokens. We have extended this process to the prefill stage. During the prefill stage, splitting the input into chunks does not impact the correctness of the computation [1]. OmniKV-prefill treats each chunk as a local observation window to filter out important tokens from full context within the "filter layers". And the **total number of selected tokens** is $r \times $ chunk\_size. Then, the "non-filter layers" use the important tokens identified by the "filter layers" as the context. For these "non-filter layers," the inputs for attention calculations consist of two parts: 1) The KV cache containing the selected important tokens. 2) The hidden states of the current chunk. Thus, the sequence length for these layers becomes $(r+1) \times $ chunk\_size. We have further added an illustration to more clearly demonstrate the process [OmniKV-prefill](https://anonymous.4open.science/r/temp_anonymous-6BDE/OmniKV-prefill-example.pdf).
>
> [1] Agrawal, Amey, et al. "Sarathi: Efficient llm inference by piggybacking decodes with chunked prefills." _arXiv preprint arXiv:2308.16369_ (2023).
>
> > What is the difference between "r=1" and the baseline?
>
> Taking the actual experimental setup of OmniKV-prefill as an example, `chunk_size=4096, r=1` means that OmniKV-prefill selects $1\times 4096$ important tokens in the "filter layers". The actual sequence length of the hidden state inputs accepted by the "non-filter layers" is thus $4096$ (important tokens)+ $4096$ (current chunk) = $8192$ tokens. When `chunk_size=4096, r=2`, the actual sequence length of the hidden state inputs accepted by the "non-filter layers" is $2 \times 4096$ (important tokens) + $4096$ (current chunk) = $12288$ tokens. This greatly reduces the input sequence length involved in the computation of the "non-filter layers", reducing the complexity of "non-filter layers" from $n^2$ to $(r+1)n$, thus accelerating the prefill stage.
>
> > The paper mentions that OmniKV “perform full attention" in "all layers" in prefilling stage, why is it faster than the baseline? Do you evict tokens in the prefilling stage in the current version?
>
> In the prefill stage, the original OmniKV performs full attention in all layers. Since prefill acceleration is also important, we extended OmniKV to the prefill stage. OmniKV-prefill only executes full attention in the "filter layers" and layers before the first "filter layer" in both prefill and decode stages. For other "non-filter layers", OmniKV-prefill performs sparse attention, which is why it's faster than the baseline. OmniKV-prefill still does not discard any tokens to maintain the model's ability in multi-step reasoning scenarios.
>
> We hope our responses address your concerns, but if there are still any aspects that remain unclear, we would be more than happy to further discuss and provide additional clarifications.

---

> ### Author Response · Authors · 2024-11-26
> **Further Explanation of OmniKV-prefill**
>
> We have further created a visual comparison to showcase the differences between OmniKV during the decode stage and OmniKV-prefill during the prefill stage. You can view the illustration via this link: [Comparison](https://anonymous.4open.science/r/temp_anonymous-6BDE/compare.pdf). This diagram also demonstrates the efficiency enhancements introduced in the prefill stage with OmniKV-prefill. Please let us know if further clarification is needed.

---

> ### Author Response · Authors · 2024-11-30
> **Response to Reviewer UJXu**
>
> Thank you for your question regarding the OmniKV integration within the lightllm framework as presented in our results. In the table labeled "Decoding latency (ms/token) on a single NVIDIA-H20," the version of OmniKV we demonstrated did not have offloading activated. Lightllm utilizes complex memory management mechanisms and currently lacks support for offloading. We are currently in the process of implementing a version that supports offloading. We will update our results and share improvements once this version is available. Thank you for your interest in our work.

---

> ### Author Response · Authors · 2024-12-02
>
> Thank you for your valuable feedback. Offloading does bring some overhead, but based on our simulation, in the scenario of long context, Lightllm + OmniKV w/ offload can still achieve **lower latency** than Lightllm + origin model and greatly expand the longest supported context. The experimental setup consists of an NVIDIA-H20 and a 16-core CPU.
>
> We consider the overhead brought by Offload from the prefill and decode stages.
>
> The prefill stage is computationally intensive, so OmniKV can use computation to cover the cost of transferring kv cache offload to CPU memory. Therefore, the additional overhead brought by the offload can be ignored, which is shown in Figure 4 where the prefill time for OmniKV with offload and the original model is almost the same.
>
> For the decode stage, there are mainly two operations that bring overhead:
>
> 1. Use the index of important tokens obtained from the "filter layer" to index the full KV cache in CPU memory.
>
> 2. Load the indexed sub KV cache to the GPU. The amount of this part of the transmission is the size of sub KV cache `num_important_tokens*num_layers_on_cpu*num_key_value_heas*head_dim*dtype_size*2`. Due to the consistency of important tokens between OmniKV layers, we can start the load of consecutive layers at the "filter layer".
>
>
> We measured the speed of CPU to GPU data transfer on server with NVIDIA H20 to be 17.7GB/s. As shown in Figure 5, `num_important_tokens=2048` is enough to achieve good results. When `num_important_tokens=2048`, the total transmission volume for outputting a token is about `2048*24*8*128*2*2/1e9=0.2GB`, which takes about 11.3ms, and the indexing operation takes about 3.5ms, so the total time is 14.8ms. It should be noted that the cpu-to-gpu bandwidth of server with A100 in Figure 4 of the paper is about 10.2GB/s, which is slower than the H20 server.
>
> When the context length is 256K (the longest context supported by an NVIDIA H20), the total serial delay is 26.4+14.8=41.2ms. But because computation can cover transfer, the time consumed by Lightllm + OmniKV w/ offload should be **less than 41.2ms**, which is still faster than the 47.1ms of Lightllm + origin model.
>
> Since the transmission volume is fixed, the latency is independent of the context length. When the GPU memory can accommodate all KV caches, OmniKV can avoid offloading. In addition, the number of layers that perform offloading can be dynamically determined based on the context length. For example, when the context length slightly exceeds the maximum that can be accommodated (e.g. 300K), we can only offload the KV cache of one layer to further reduce the impact of offloading on efficiency.
>
> We hope our responses address your concerns.

---

> > ### Comment · Reviewer_UJXu · 2024-12-02
> >
> > Thank you for the response. I think most of my concerns have been addressed. Hope to see that the latency numbers (e.g., Figure 4) in the original submission can be updated to the latest implementation. I will raise my score to 6.

---

> > > ### Author Response · Authors · 2024-12-03
> > >
> > > We are pleased to hear that our responses have addressed most of your concerns. Your suggestions are immensely valuable to us. We will update Figure 4 in our next submission to include the latest latency numbers reflecting our most recent implementation. We are committed to continuously refining our paper. Thank you once again for your insightful comments and for raising your score.

---

### Official Review · Reviewer_eZxa · 2024-11-04

**Soundness:** 3
**Presentation:** 3
**Contribution:** 2
**Rating:** 6
**Confidence:** 5

**Summary:**

The paper proposes an inference optimization framework which leverage inter-layer attention similarity to offload most of the KV into CPU and select relevant context dynamically to alleviate the memory bound on GPU while maintaining the integrity of contexts.

**Strengths:**

1. The method can keep full context, which was reflected in the needle and LongBench tasks.
2. The method can achieve preserve better model quality while having more speed-up over other methods in this class.

**Weaknesses:**

1. The memory manipulation doesn't seem to be compatible with fundamental serving mechanisms like continuous batching and prefix caching. It's questionable how such algorithm can work in real world.
2. It will be quite challenging to make the proposed method work with tensor parallel,pipeline parallel, and context parallel, which are also essential in production.

**Questions:**

1. Related to weakness 1, how does this method compare to vLLM/SGLang with FlashInfern/FlashAttention backend, in terms of throughput and latency? If slower, is OmniKV compatible with practical serving framework like vLLM?
2. Related to weakness 2, can you benchmark TP=4,8 for <10b scale mdoels? Also, tp=8, pp=4, for any 70B models? I'd like to see (1) latency and throughput for prompt length of 128k, 256k, 512k(probably need ctx parallel) and output length of 512, (2) quality test on needle-in-a-haystack on 128k, 256k, 512k ctx.
3. Could you benchmark against [1], for both speed and quality preserving?

[1] MInference 1.0: Accelerating Pre-filling for Long-Context LLMs via Dynamic Sparse Attention

---

> ### Author Response · Authors · 2024-11-21
> **Response (1/3) to Reviewer eZxa**
>
> > Weakness 1: The memory manipulation doesn't seem to be compatible with fundamental serving mechanisms like continuous batching and prefix caching. It's questionable how such algorithm can work in real world.
>
> OmniKV is **compatible with existing inference frameworks**. We have implemented OmniKV within [Lightllm](https://github.com/ModelTC/lightllm). As Lightllm is a lightweight inference framework which is easier to adapt compared to popular vLLM. The development on vLLM is still ongoing.
>
> Since Lightllm supports continuous batching and prefix caching, we compared the impacts of enabling and disabling these features on inference efficiency. Experiments here were conducted on **a single NVIDIA-H20**.
>
> **Continuous Batching:**
>
> We first sending a 64K-length request (A), followed by a 30-second wait before sending two 4K-length requests (B and C). During the process, the observed real-time running batch size changed as follows:
>
> `Batch Size: 0 => 1 (A start) => 3 (B, C join) => 2 (A end, response) => 0 (B, C end, response)`
>
> Enabling Continuous Batching reduced the end-to-end latency for requests B and C from **28.9** seconds to **23.5** seconds. But it also led to a slight increase in the latency of request A (**37.8s** => **41.7s**), as A, B, and C were decoding simultaneously for a certain period.
>
> **Prefix Caching:**
>
> Using the novel *Harry Potter*, we initially sent a request to summarize the entire book. Then, after randomly removing a portion of the latter content, we sent these sub-contexts for summarization. In this scenario, the overall average end-to-end latency was reduced from **65.8** seconds to **35.49** seconds.
>
> **Algorithmic compatibility:** OmniKV utilizes the exact same decoding process as standard LLMs, with the only distinction being that in a few "filter layers". These layers mask out unimportant caches but retain all KV caches. So, OmniKV remains fully compatible with continuous batching and prefix caching.
>
>
>
> > Related to weakness 1, how does this method compare to vLLM/SGLang with FlashInfern/FlashAttention backend, in terms of throughput and latency? If slower, is OmniKV compatible with practical serving framework like vLLM?
>
>
>
> The experimental results are as follows. All experiments were conducted on **a single NVIDIA-H20**. "origin" refers to using original model.
>
> **The throughput results of decoding (tp=1, batch size=1):**
>
> | Exp. settings | Lightllm+OmniKV | Lightllm+origin | vLLM+origin |
> |:------------- | --------------- | --------------- | ----------- |
> | 16K           | 54.1            | 51.1            | **72.4**    |
> | 32K           | 53.3            | 46.7            | **62.4**    |
> | 64K           | **52.8**        | 40.1            | 50.4        |
> | 128K          | **45.3**        | 31.0            | 36.0        |
> | 256K          | **37.9**        | 21.2            | 23.1        |
>
> Due to OmniKV's reduction in computation, Lightllm+OmniKV achieved the best results when the sequence length exceeded 64K. Moreover, OmniKV achieved a 1.64x speedup for an input length of 256k.

---

> ### Author Response · Authors · 2024-11-21
> **Response (2/3) to Reviewer eZxa**
>
> > It will be quite challenging to make the proposed method work with tensor parallel,pipeline parallel, and context parallel, which are also essential in production.
> >
> > can you benchmark TP=4,8 for <10b scale models? Also, tp=8, pp=4, for any 70B models?
> >
> > I'd like to see (1) latency and throughput for prompt length of 128k, 256k, 512k(probably need ctx parallel) and output length of 512
>
>
>
> OmniKV is **compatible with these three parallelism methods**. Due to the simplicity of OmniKV, the adaptation process is relatively straightforward.
>
> **Tensor parallelism:** We have implemented OmniKV w/ tensor parallelism in Lightllm. Experimental results indicate that with TP enabled, OmniKV achieves considerable acceleration compared to vLLM with original model.
>
> **Throughput results of decoding (tp=4), using Llama-3-8B-Instruct-262K**:
>
> | Exp. settings | Lightllm+OmniKV | Lightllm+origin | vLLM+origin |
> | ------------- | --------------- | --------------- | ----------- |
> | 128k, bs=16   | **345.9**       | 217.7           | 221.3       |
> | 256k, bs=8    | **177.5**       | 106.1           | 109.4       |
> | 512k, bs=4    | **89.1**        | 51.2            | 52.9        |
>
> **Average latency of per token for each request (ms/token, tp=4):**
>
> | Exp. settings | Lightllm+OmniKV | Lightllm+origin | vLLM+origin |
> | ------------- | --------------- | --------------- | ----------- |
> | 128k, bs=16   | **46.2**        | 73.5            | 72.3        |
> | 256k, bs=8    | **44.9**        | 75.4            | 73.1        |
> | 512k, bs=4    | **44.9**        | 78.1            | 75.6        |
>
> **Throughput results of decoding (tp=8), using Llama-3-8B-Instruct-262K:**
>
> | Exp. settings | Lightllm+OmniKV | Lightllm+origin | vLLM+origin |
> | ------------- | --------------- | --------------- | ----------- |
> | 128k, bs=32   | **566.9**       | 409.8           | 407.1       |
> | 256k, bs=16   | **306.8**       | 204.6           | 199.4       |
> | 512k, bs=8    | **155.0**       | 98.5            | 95.4        |
>
> **Average latency of per token for each request (ms/token, tp=8):**
>
> | Exp. settings | Lightllm+OmniKV | Lightllm+origin | vLLM+origin |
> | ------------- | --------------- | --------------- | ----------- |
> | 128k, bs=32   | **56.4**        | 78.1            | 78.6        |
> | 256k, bs=16   | **52.1**        | 78.2            | 80.2        |
> | 512k, bs=8    | **51.6**        | 81.2            | 83.8        |
>
> Lightllm+OmniKV experiences reduced decoding latency as sequence length increases. This is because the "non-filter layers" have a fixed sequence length of 2048. As the batch size decreases, the computation also decreases for "non-filter layers". But when context length decreases, the computation remains the same for "non-filter layers".
>
> **Technical detail of tp:** The attention heads is distributed across different GPUs in tensor parallelism, and the outputs are aggregated using all-reduce, making inter-GPU communication a key factor in efficiency.
>
> For $\text{tp}=n$ , OmniKV maintains $n$ indices of important tokens for each card. Each index of important tokens is computed solely by the card's own attention heads, thereby eliminating the need for inter-GPU communication. Since different attention heads may have distinct indices, the system intuitively exhibits greater flexibility. We have also substantiated this through experiments.
>
> | Exp. settings | 2WikiMQA | Qasper   | HotpotQA | lcc      | qmsum    |
> | ------------- | -------- | -------- | -------- | -------- | -------- |
> | tp=8          | 21.7     | 28.0     | **27.3** | **52.3** | **25.9** |
> | tp=1          | **21.8** | **28.6** | 27.0     | 51.4     | 24.9     |
>
> **Pipeline parallelism:** We have implemented OmniKV w/ tensor parallelism in transformers (huggingface). The experiments of Llama-3.1-70B shown in Table 1 is conducted using pipeline parallelism. And the support for Lightllm and vLLM is ongoing.
>
> To implement OmniKV in pipeline parallelism, we only need to load the corresponding subset of KV cache onto the correct GPU device. The overhead introduced is quite small, usually `batch_size * k_size * tp_num`, with `k_size` usually set to 2048.
>
> **Context parallelism:** OmniKV remains fully compatible. We typically require only around 2048 tokens to achieve excellent results. For this short sequence length, the communication latency in multi-GPU parallelism often exceeds the computational latency of a single GPU. Hence, we can employ context parallelism only in dense (full) attention layers, while utilizing a single GPU for sparse attention layers. This means we simply need to configure which layers will implement context parallelism. We can allocate samples to different GPUs when the batch size > 1, to prevent resource wastage.
>
>
>
> Regarding the 70B model setup with tp=8 and pp=4, we currently lack of GPUs to run the experiments. We are actively reaching out to acquire the resources and will provide the results once available.

---

> ### Author Response · Authors · 2024-11-21
> **Response (3/3) to Reviewer eZxa**
>
> > (2) **quality test on needle-in-a-haystack** on 128k, 256k, 512k ctx.
>
> Utilizing the model Llama-3-8B-Instruct-Gradient-1048k, OmniKV achieved perfect accuracy on the needle-in-a-haystack benchmark. We have integrated the results of the needle-in-a-haystack experiment with a context length of up to 512k into Figure 7 of the paper.
>
>
>
> > Could you benchmark against [1], for both speed and quality preserving?
>
> Firstly, it is crucial to clarify that MInference is a method specifically designed to optimize **prefill** latency, whereas OmniKV targets the latency in the **decoding** phase and extends the maximum context length supported under memory constraints. These two methods are orthogonal to one another. However, we have still applied OmniKV to optimize the prefill stage, naming this approach **OmniKV-Prefill**.
>
> Specifically, during the prefill stage, we divide the sequence dimension into multiple chunks for computation. For each chunk, we also use the "filter layer" to select important tokens. Since the sequence length of the query chunk is no longer 1, we employ the uniform context selector introduced in Section 4.2 to select important tokens in the "filter layer".
>
> Here, "r" denotes the size of the selected tokens, with the ratio of token size to chunk size being "r". Thus, the larger the value of r, the less information is lost.
>
> **Prefill latency on a single NVIDIA-H20:**
>
> | Exp. settings           | 256k  | 128k | 64k  | 32k | 16k |
> | ----------------------- | ----- | ---- | ---- | --- | --- |
> | OmniKV-prefill, r=1     | 105.9 | 36.7 | 14.3 | 6.2 | 2.8 |
> | **OmniKV-prefill, r=2** | 107.2 | 38.8 | 15.9 | 6.5 | 2.9 |
> | OmniKV-prefill, r=4     | 112.9 | 42.8 | 17.1 | 6.8 | 3.0 |
> | OmniKV-prefill, r=8     | 133.3 | 49.2 | 19.0 | 6.9 | 2.9 |
> | MInference              | 61.6  | 28.1 | 13.0 | 6.5 | 3.7 |
> | Origin                  | 203.8 | 58.6 | 18.3 | 6.5 | 2.6 |
>
> MInference is specifically designed for **Prefill** acceleration and optimizes multiple CUDA kernels. But OmniKV-prefill solely relies on the Huggingface-transformers library, consequently less efficient. However, OmniKV-prefill still achieves a **1.90**x latency improvement over the original model with an input length of 256K.
>
> **Quality Preserving:**
>
> | Exp. settings           | %Mem | Single-Doc QA | Multi-Doc QA | Summarization | Few-Shot | Syntheic Tasks | Code     |
> | ----------------------- | ---- | ------------- | ------------ | ------------- | -------- | -------------- | -------- |
> | Llama-3-8B-262K         | 100% | 29.2          | 22.9         | 24.9          | 65.9     | 43.5           | 48.9     |
> | Minference              | 30%  | 29.2          | 22.4         | **25.3**      | 65.8     | 41.0           | 49.8     |
> | OmniKV-prefill, r=1     | 30%  | 29.8          | 22.0         | 24.0          | 63.5     | 33.3           | **50.0** |
> | **OmniKV-prefill, r=2** | 30%  | **30.0**      | 23.1         | 24.3          | 65.1     | 40.0           | 49.6     |
> | OmniKV-prefill, r=4     | 30%  | 29.8          | 22.9         | 24.3          | 65.9     | 42.0           | 49.1     |
> | OmniKV-prefill, r=8     | 30%  | 29.8          | 22.9         | 24.3          | **65.9** | 42.0           | 49.0     |
> | OmniKV-prefill, all     | 30%  | 29.6          | **23.3**     | 23.7          | 64.0     | 41.5           | 48.4     |
>
> The results indicate that r=2 has achieved a fine balance between efficiency and performance.

---

> ### Author Response · Authors · 2024-11-25
> **Response to Reviewer eZxa**
>
> Dear Reviewer eZxa,
>
> Thank you for taking the time to provide us with your thoughtful and constructive feedback. We greatly appreciate your insights and have prepared our rebuttal to address each of your concerns with the utmost respect.
>
> We kindly ask if you could confirm whether our responses have sufficiently resolved your concerns. Please feel free to reach out with any further questions or suggestions, as we remain open to continued discussion.
>
> Best regards,  The Authors

---

> ### Author Response · Authors · 2024-11-29
>
> Dear Reviewer eZxa,
>
> We hope this message finds you well. Thank you once again for your insightful review and the valuable feedback provided. We have carefully addressed the concerns you raised in our rebuttal, particularly focusing on the deployment challenges of OmniKV in practical applications and comparison with MInference.
>
> We demonstrated that OmniKV, when integrated with Lightllm (a lightweight inference acceleration framework), enhances efficiency, showing that Lightllm+OmniKV outperforms vLLM+origin and Lightllm+origin configurations. Additionally, we have extended OmniKV to the prefill stages, referring to this adaptation as OmniKV-prefill. This adaptation has achieved a significant speed improvement of 1.90x compared to the original model, while maintaining performance quality on par with the MInference benchmarks.
>
> We sincerely hope that our responses and the additional benchmarks provided have addressed your concerns effectively. We are keen to receive your feedback to confirm whether the issues have been resolved to your satisfaction or if further clarification is needed.
>
> Thank you for your continued engagement and support in refining our work. We look forward to your response.

---

> > ### Comment · Reviewer_eZxa · 2024-11-30
> > **Thank you for your response**
> >
> > Thank you very much for the clarification. I appreciate the author's efforts in integrating it into vLLM and measuring continuous batching. I have raised my score.

---

> > > ### Author Response · Authors · 2024-12-01
> > >
> > > Thank you for your insightful and positive feedback, and for raising the score. We greatly appreciate your valuable support and recognition.

---

### Official Review · Reviewer_ztbP · 2024-11-04

**Soundness:** 3
**Presentation:** 3
**Contribution:** 2
**Rating:** 6
**Confidence:** 4

**Summary:**

This paper introduces a KV cache selection algorithm based on cross-layer similarity and shows reasonable downstream performance and wall clock speedups.

**Strengths:**

This paper explains the cross-layer KV cache selection similarity well and designs a real-world system based on this. The evaluations include the 70B-level model's results, and the decoding speed is also very impressive.

**Weaknesses:**

1 Inference-time pipeline parallelism, especially in the same node, is uncommon.   "which is 1.87 times faster than the original model running on three A100 GPUs in pipeline" does not make sense to me.  A tensor parallelism baseline is expected in this case, even if I can understand it is very hard to beat.
2 This paper uses cross-layer KV cache selection similarity. However, this idea was introduced by InfiniGen in a previous paper. No proper discussion is seen in this paper. If this is solved, I will raise my score.
3 The baselines shown in this paper are not proper. Only InfLLM is a dynamic KV selection method, but it is mainly used for extending context windows. I will ask for comparisons with Quest or Loki, as they are also dynamic KV selection methods. Static KV selection methods like H2O can not only save inference time but also save total memory without CPU offloading. If this is solved, I will raise my score.

[1] Lee, W., Lee, J., Seo, J., & Sim, J. (2024). {InfiniGen}: Efficient generative inference of large language models with dynamic {KV} cache management. In 18th USENIX Symposium on Operating Systems Design and Implementation (OSDI 24) (pp. 155-172).
[2] Tang, J., Zhao, Y., Zhu, K., Xiao, G., Kasikci, B., & Han, S. (2024). Quest: Query-Aware Sparsity for Efficient Long-Context LLM Inference. arXiv preprint arXiv:2406.10774.
[3] Singhania, P., Singh, S., He, S., Feizi, S., & Bhatele, A. (2024). Loki: Low-Rank Keys for Efficient Sparse Attention. arXiv preprint arXiv:2406.02542.

**Questions:**

As stated in weakness.

---

> ### Author Response · Authors · 2024-11-21
> **Response (1/2) to Reviewer ztbP**
>
> > Inference-time pipeline parallelism, especially in the same node, is uncommon. A tensor parallelism baseline is expected in this case, even if I can understand it is very hard to beat.
>
> Sorry for the confusing description. We employed pipeline parallelism solely to ensure a **fair comparison in computation** with the original model. We then implement OmniKV with tensor parallelism on Lightllm, and test the throughput of decoding shown below. Here "origin" means using original model.
>
> **Throughput results of decoding (tp = 4), measured in tokens per second, using Llama-3-8B:**
>
> | Exp. settings | Lightllm+OmniKV | Lightllm+origin | vLLM+origin |
> | ------------- | --------------- | --------------- | ----------- |
> | 16k, bs=100   | **1450.5**      | 1418.0          | 1395.2      |
> | 32k, bs=64    | **1104.3**      | 803.7           | 802.1       |
> | 64k, bs=32    | **627.0**       | 414.6           | 440.5       |
> | 128k, bs=16   | **345.9**       | 217.7           | 221.3       |
> | 256k, bs=8    | **177.5**       | 106.1           | 109.4       |
> | 512k, bs=4    | **89.1**        | 51.2            | 52.9        |
>
> It is worth noting that the half-precision compute power of the NVIDIA H20 is **148** TFLOPS, which is lower than the **312** TFLOPS of the A100. If you have the throughput results for the A100 for comparison.
>
> The experimental results demonstrate that OmniKV is fully compatible with tensor parallelism and yields superior performance compared to the original model on popular vLLM.
>
>
>
> > This paper uses cross-layer KV cache selection similarity. However, this idea was introduced by InfiniGen in a previous paper. No proper discussion is seen in this paper. If this is solved, I will raise my score.
>
> InfiniGen introduced cross-layer similarity between **consecutive two layers**, leveraging this characteristic to pre-select the critical KV cache for layer $i+1$ during the processing of layer $i$.
>
> However, OmniKV has uncovered that certain "filter layers" can identify tokens with consistently high attention scores **across multiple consecutive layers**. Although InfiniGen loads the KV cache for layer $i+1$ while processing layer $i$, the loading time may still exceed the computation time of layer $i$, resulting in GPU idling. In contrast, OmniKV can pre-load the cache for multiple layers $[i+2, i+n],(n>2)$ during the "filter layer" $i$, thus significantly reducing GPU idle time. This is also why we designate layer $i+1$ to perform full attention when layer $i$ is "filter layer".
>
>
>
> > The baselines shown in this paper are not proper. Only InfLLM is a dynamic KV selection method, but it is mainly used for extending context windows. I will ask for comparisons with Quest or Loki, as they are also dynamic KV selection methods.
>
> Thank you for your advice. We conducted experiments with Quest on LongBench. We utilized the model LongChat-v1.5-7b-32k, as Quest does not support GQA. In these experiments, both Quest and OmniKV were assigned a cache budget corresponding to 30% of the total sequence length.
>
> **Performance of longchat-v1.5-7b-32k on LongBench**
>
> | Exp. settings        | %Mem | Qasper   | HotpotQA | GovReport | TriviaQA | NarrativeQA | MultifieldQA |
> | -------------------- | ---- | -------- | -------- | --------- | -------- | ----------- | ------------ |
> | longchat-v1.5-7b-32k | 100% | 29.4     | 32.0     | 30.6      | 81.2     | 20.7        | 43.8         |
> | Quest                | 30%  | 31.7     | 32.2     | **30.2**  | **84.3** | 20.1        | **42.0**     |
> | OmniKV               | 30%  | **31.9** | **33.4** | 29.6      | 83.5     | **21.5**    | 40.9         |
>
> The experimental results indicate that OmniKV achieves performance comparable to Quest, while additionally reducing KV cache memory usage by 70%.
>
> **Decoding efficiency (batch size = 1), measured in milliseconds per token:**
>
> | Exp. settings   | 120k     | 96k      | 64k      | 32k      | 16k      |
> | --------------- | -------- | -------- | -------- | -------- | -------- |
> | Lightllm+origin | 35.6     | 30.6     | 27.3     | 21.4     | 19.6     |
> | Quest           | OOM      | OOM      | OOM      | **16.2** | **15.4** |
> | Lightllm+OmniKV | **23.6** | **22.3** | **21.2** | 20.1     | 18.7     |
>
> The experimental setup was conducted on a single NVIDIA-H20, utilizing the LongChat-v1.5-7b-32k model. The results indicate that OmniKV has comparable efficiency with Quest in short-text scenarios. However, OmniKV is capable of supporting significantly longer texts.
>
> The performance of OmniKV in short-text scenarios still holds potential for further improvement. Quest is implemented using customized kernels, whereas OmniKV lacks deeply optimized CUDA kernels.

---

> ### Author Response · Authors · 2024-11-21
> **Response (2/2) to Reviewer ztbP**
>
> > Static KV selection methods like H2O can not only save inference time but also save total memory without CPU offloading. If this is solved, I will raise my score.
>
> The static KV selection methods like H2O are susceptible to discarding tokens that may not have garnered sufficiently high attention at earlier stages, but these tokens could prove pivotal in later phases of generation. During multi-step inference (e.g., in CoT scenarios), tokens receiving high attention scores shift as generation evolves as shown in Figure 1b. Therefore, OmniKV retains the entire KV cache and dynamically selects during the decoding process.
>
> The comparison between OmniKV and H2O is presented in Tables 1 and 2. OmniKV maintains higher quality than H2O, particularly in Chain-of-Thought scenarios, as illustrated in Figure 3 and Table 6.

---

> ### Comment · Reviewer_ztbP · 2024-11-22
>
> I do not think offloading the KV cache to the CPU can be seen as saving memory. I suggest the authors to correct this. Offloading is more like a system technique rather than actually reducing memory. Other than this, the authors cite and discuss InfiniGen in related works and provide a comparison with Quest, which is a more state-of-the-art and related baseline. This paper also seems to have a strong system implementation. I raise my score to 6.

---

> > ### Author Response · Authors · 2024-11-24
> > **Response to Reviewer ztbP**
> >
> > Thank you for your valuable feedback and for raising your score. We appreciate your suggestion regarding the distinction between offloading and memory reduction, and we have revised our wording in the new version to reflect this more accurately. We're also glad you recognized the strength of our system implementation and our comparison with state-of-the-art baselines. Thank you again for your thoughtful comments!

---

### Official Review · Reviewer_diVc · 2024-11-05

**Soundness:** 3
**Presentation:** 3
**Contribution:** 3
**Rating:** 6
**Confidence:** 2

**Summary:**

The inference of large language models (LLM) is memory intensive. This paper proposes a new method for reducing the GPU memory usage when doing LLM inference on GPUs. The main insight of this paper is the similarity of attention score matrix between different consecutive layers. By leveraging this insight, the author proposes a new method to reduce the GPU memory usage of the KV cache  by over 75% via offloading to CPU memory, while not significantly affect the accuracy.

**Strengths:**

* The author aims at a timely problem of reducing memory usage of KV cache for LLM inference.
* The proposed method is effect yet simple to implement.
* The author demonstrate real performance gain and capability to run LLM inference with longer sequence length with real workloads.

**Weaknesses:**

* There is no mathematical guarentee that the propose method is universally applicable to all LLMs.

**Questions:**

Thanks for submitting the marvelous paper to ICLR. In general, I believe the paper is trying to solve a timely and important problem. While I appreciate the nice results the proposed methods can achieve, my major concern of this paper is that it introduces some hyper-parameters that could be non-trivial to tune. So, I am wondering how would you tune the filter layers in your designed algorithm?

Besides, since the author does not provide any intuition or mathematical guarantee in the paper, I would be concerned about the generalizability to bigger LLMs such as Llama 3.1 405B. So I am wondering if you have some intuition to explain why attention score matrix could be similar across different layers? While I understand it could be expensive to run experiments with larger LLMs, do you have any intuition that the proposed method will work even with larger LLMs?

---

> ### Author Response · Authors · 2024-11-21
> **Response to Reviewer diVc**
>
> > Is it introduces some hyper-parameters that could be non-trivial to tune?
>
> Firstly, OmniKV is insensitivity to hyperparameters, as shown in Table 8 of the paper, where the majority of parameter configurations yield similar results. The only exception is when the fourth layer is used as a "filter layer," which leads to relatively poor performance. However, this phenomenon aligns with the lower Inter-Layer Attention Similarity of layer 4 depicted in the left graph of Figure 6a.
>
> Furthermore, as shown in Figures 5 and 6, we find that the "filter layers" are an inherent property of the model and do not vary with changes in tasks. This implies that for a given model, we only need to set one hyperparameter that can be applied across different tasks, greatly facilitating OmniKV's application in real-world scenarios.
>
> > Why attention score matrix could be similar across different layers?
> >
> > The generalizability to bigger LLMs such as Llama 3.1 405B.
>
> Deja Vu[1] have noted the existence of hidden state similarity between consecutive two layers, and posited that this may primarily arise from residual connections.
>
> We intuitively believe that the crucial information carried by certain tokens should be progressively deepened and refined across layers, which may cause multiple layers to focus on a repeated subset of tokens. Thus, further experiments explored the similarity of attention maps between arbitrary layers. As shown in Figure 8, even between layers that are far apart, there exists a high degree of Inter-Layer Attention Similarity, providing some validation for this hypothesis.
>
> In Llama 3.1 405B, as illustrated in Figure 9, the Inter-Layer Attention Similarity validates our intuition. We then ensure this through experiments on 2 tasks from LongBench.
>
> | Exp. settings  | Qasper | Qmsum |
> | -------------- | ------ | ----- |
> | Llama 3.1 405B | 50.0   | 25.5  |
> | OmniKV         | 48.5   | 25.9  |
>
> [1] Liu, Zichang, et al. "Deja vu: Contextual sparsity for efficient llms at inference time." _International Conference on Machine Learning_. PMLR, 2023.

---

> > ### Comment · Reviewer_diVc · 2024-11-26
> >
> > I would like to appreciate the authors for their response. I think the response has generally answered my questions. To even make the paper stronger, I would like to suggest the author to talk the generalizability to larger LLMs in the paper.

---

> > > ### Author Response · Authors · 2024-11-26
> > > **Response to Reviewer diVc**
> > >
> > > Thank you very much for your thoughtful suggestion. We have added a discussion on the generalizability to larger models (i.e., a 405B parameter model) in Section 5 of the main text. Additionally, we have included detailed explanations and supporting experiments in Appendix D.8. We hope this further strengthens the paper.

---

### Author Response · Authors · 2024-11-21
**General Response**

Dear reviewers,

We extend our deepest gratitude to all the reviewers for your insightful and constructive comments. We have addressed each of your concerns individually under the corresponding reviews.

We have revised the paper and highlighted the key changes to yellow. The revised part mainly includes:

- New Section (D.8): Effectiveness of OmniKV on larger Models (405B).

- New Section (D.9): Compatibility of OmniKV to existing inference techniques and corresponding experiments.

- New Section (D.10): Prefill acceleration.

- Comparison with InfiniGen in related work.

- Modified confusing sentences.

We will continue refining the paper's structure and improving its content quality. Additionally, we have identified some common issues and presented the results below.

### The comparison of efficiency between OmniKV and original model utilizing tensor parallelism (Reviewer ztbP, eZxa, UJXu)

We have made OmniKV compatible with tensor parallelism, and OmniKV achieves superior results compared to the original model on vLLM. Please note that all experiments conducted during the rebuttal phase were performed using NVIDIA H20 GPUs. For comparison, the half-precision computational capabilities of the H20 and A100 GPUs are 148 TFLOPS and 312 TFLOPS, respectively.

**Throughput results of decoding (tp = 4), measured in tokens per second, using Llama-3-8B:**

| Exp. settings | Lightllm+OmniKV | Lightllm+origin | vLLM+origin |
| ------------- | --------------- | --------------- | ----------- |
| 16k, bs=100   | **1450.5**      | 1418.0          | 1395.2      |
| 32k, bs=64    | **1104.3**      | 803.7           | 802.1       |
| 64k, bs=32    | **627.0**       | 414.6           | 440.5       |
| 128k, bs=16   | **345.9**       | 217.7           | 221.3       |
| 256k, bs=8    | **177.5**       | 106.1           | 109.4       |
| 512k, bs=4    | **89.1**        | 51.2            | 52.9        |

### OmniKV's compatibility with serving mechanisms and parallelism methods (Reviewer ztbP, eZxa, UJXu)

OmniKV is **compatible with serving mechanisms and parallelism methods**. We have adapted OmniKV to the lightweight inference framework [Lightllm](https://github.com/ModelTC/lightllm), leveraging this framework to support Continuous Batching, Prefix Caching, and Tensor Parallelism. We have also implemented support for pipeline parallelism in the Huggingface Transformers version. The development on vLLM is still ongoing. We will incorporate it after obtaining the results. The throughput performance of decoding after applying OmniKV on Lightllm generally follows this pattern: **Lightllm+OmniKV > vLLM+origin > Lightllm+origin**.

### Prefill Acceleration (Reviewer eZxa,UJXu)

One of main targets of OmniKV is accelerate **decoding** speed. Meanwhile, given the importance of prefill acceleration, we have modified OmniKV to optimize the prefill stage as well. Experimental results indicate that OmniKV-prefill achieves a **1.90**x reduction in latency without performance loss compared to the original model with an input length of 256K. The code is implemented using the Transformers library with Flash Attention.

---

### Meta-Review · Area_Chair_AvVY · 2024-12-20

**Metareview:**

The paper presents a method that exploits inter-layer attention similarity to efficiently reduce KV cache memory usage in LLM inference while maintaining model performance, enabling significantly longer context lengths on limited GPU hardware.
Reviewers after the author responses were largely convinced of the merits of the paper, as authors added additional demonstrations of  integration with inference frameworks and benchmark results.

We rely on the authors to incorporate the discussion points in the final version.

**Additional Comments On Reviewer Discussion:**

The author feedback phase was productive, converging to good agreement between all parties

---

### Decision · Program_Chairs · 2025-01-22

Accept (Poster)